# Incorporating regulatory interactions into gene-set analyses for GWAS data: A controlled analysis with the MAGMA tool

**David Groenewoud**, **Avinoam Shye**, **Ran Elkon***

Department of Human Molecular Genetics and Biochemistry, Sackler Faculty of Medicine, Tel Aviv University, Ramat Aviv, Israel

* ranel@tauex.tau.ac.il

## Abstract

To date, genome-wide association studies have identified thousands of statistically-significant associations between genetic variants, and phenotypes related to a myriad of traits and diseases. A key goal for human-genetics research is to translate these associations into functional mechanisms. Popular gene-set analysis tools, like MAGMA, map variants to genes they might affect, and then integrate genome-wide association study data (that is, variant-level associations for a phenotype) to score genes for association with a phenotype. Gene scores are subsequently used in competitive gene-set analyses to identify biological processes that are enriched for phenotype association. By default, variants are mapped to genes in their proximity. However, many variants that affect phenotypes are thought to act at regulatory elements, which can be hundreds of kilobases away from their target genes. Thus, we explored the idea of augmenting a proximity-based mapping scheme with publicly-available datasets of regulatory interactions. We used MAGMA to analyze genome-wide association study data for ten different phenotypes, and evaluated the effects of augmentation by comparing numbers, and identities, of genes and gene sets detected as statistically significant between mappings. We detected several pitfalls and confounders of such "augmented analyses", and introduced ways to control for them. Using these controls, we demonstrated that augmentation with datasets of regulatory interactions only occasionally strengthened the enrichment for phenotype association amongst (biologically-relevant) gene sets for different phenotypes. Still, in such cases, genes and regulatory elements responsible for the improvement could be pinpointed. For instance, using brain regulatory-interactions for augmentation, we were able to implicate two acetylcholine receptor subunits involved in post-synaptic chemical transmission, namely *CHRNB2* and *CHRNE*, in schizophrenia. Collectively, our study presents a critical approach for integrating regulatory interactions into gene-set analyses for genome-wide association study data, by introducing various controls to distinguish genuine results from spurious discoveries.

**Data Availability Statement:** Datasets of RIs and GWAS summary statistics used in our work can be downloaded from their original sources (refer to references listed in Tables 1 and 2 and in S1 Table).

We provide all data necessary to replicate our evaluations of gene scores and gene-set scores, namely: (a) unadjusted gene scores (genuine scores: https://doi.org/10.6084/m9.figshare.14981151.v1) (EPVP scores: https://doi.org/10.6084/m9.figshare.14980452.v1), adjusted gene scores (genuine scores: https://doi.org/10.6084/m9.figshare.14986929.v1) (EPVP scores: https://doi.org/10.6084/m9.figshare.14986956.v1), and gene-set scores (genuine scores: https://doi.org/10.6084/m9.figshare.14986941.v1) (EPVP scores: https://doi.org/10.6084/m9.figshare.14986962.v1) (b) input to, and output from, "rrvgo" for elimination of redundant, significant gene sets (https://doi.org/10.6084/m9.figshare.14986974.v1) (c) output from IRED for testing the robustness of a gain at the level of a gene set (https://doi.org/10.6084/m9.figshare.15000444.v1) (d) coverage (https://doi.org/10.6084/m9.figshare.14986986.v1). We provide our SNV-to-gene mappings for those wishing to use these mappings to execute gene scoring and gene-set analysis for themselves with MAGMA (https://doi.org/10.6084/m9.figshare.14979780.v1). Our mappings contain sufficient information to define intragenic and extragenic SNVs according to our baseline model, for those wishing to use these mappings to replicate the EPVP permutation control. Regarding this point, we provide the background set of 7,398,358 SNVs represented in the European reference-population of the 1000-Genomes Project for filtering summary statistics (https://doi.org/10.6084/m9.figshare.14988114.v1). We supply an R script that executes MAGMA gene-scoring and gene-set analysis for two user-supplied SNV-to-gene mappings, and then runs EPVP as a control strategy: https://github.com/dgroenewoud/AUG-MAGMA.

**Funding:** RE is supported by grants from the German-Israeli Project DFG RE 4193/1-1, from the Israel Science Foundation (ISF grant No. 2118/19), and from the Koret-UC Berkeley-Tel Aviv University (KBT) Initiative in Computational Biology and Bioinformatics (KBT-TAU-2018.1). DG and AS were supported in part by fellowships from the Edmond J. Safra Center for Bioinformatics at Tel Aviv University. RE is a Faculty Fellow of the Edmond J. Safra Center for Bioinformatics at Tel Aviv University. The funders had no role in study design, data collection and analysis, decision to publish, or preparation of the manuscript.

**Competing interests:** The authors have declared that no competing interests exist.

## Author summary

Genome-wide association studies have identified thousands of statistically-significant associations between genetic variants and diverse phenotypes, but translating these associations into functional mechanisms remains a major challenge. Gene-set analyses for genome-wide association study data, which consider the signal contained in practically all variants tested for association with a phenotype, seek functionally-annotated collections of genes that are enriched for phenotype association. To do so, variants must first be mapped to genes they are thought to affect. Usually, variants are mapped to genes in their proximity, even if it is believed that the effects of many variants are, in fact, mediated via long-range regulatory interactions. Here, we demonstrate that augmenting a proximity-based mapping scheme with datasets of regulatory interactions, can strengthen the enrichment for phenotype association amongst gene sets that are biologically-relevant to phenotypes. Yet, by implementing controls, we reveal that apparent improvements are often trivial, or can be explained by confounding. We discuss some examples of meaningful improvements, and pinpoint genes and regulatory elements responsible for them. Incorporating regulatory interactions into gene-set analyses for genome-wide association study data is expected to become the norm, and our work outlines a critical approach for doing so.

## Introduction

Over the past fifteen years, genome-wide association studies (GWASs) have identified thousands of statistically-significant associations between single-nucleotide variants (SNVs) and phenotypes linked to a myriad of traits and diseases [1]. Aiming to uncover their molecular foundations, a key goal for human-genetics research is to translate these associations into functional mechanisms. This involves using SNV-level associations to pinpoint phenotype-modifying genes and biological processes [2]. Though linkage disequilibrium (LD) and untyped genetic variation complicate this procedure by hindering the identification of causal variants per se [3], it is now recognized that significantly-associated SNVs tend to implicate sequence variation within non-coding regulatory elements like enhancers, as opposed to within protein-coding sequences [4–11]. Such sequence variation is thought to alter transcription-factor binding, with repercussions for the regulation of target genes [12]. Knowledge of regulatory elements and the genes they regulate is thus important to functional analyses for GWAS data.

Conventionally, functional analyses for GWAS data focus exclusively on interpreting significantly-associated SNVs. To limit false positives, these are defined by a stringent statistical threshold (usually $p \leq 5 \times 10^{-8}$, corresponding to Bonferroni's correction for ~$10^6$ tests) [13,14]. This makes their detection challenging, because individual SNVs usually have very small effects on phenotypic variation [4,15]. In fact, many true associations with phenotypes are thought to be undetectable with realistic cohort sizes [16,17]. Restricting analyses to statistically-significant SNVs is thus expected to limit functional interpretation. This is corroborated by the observation that the proportion of phenotypic variation explained by SNVs (that is, SNV-level heritability), is often boosted considerably by accounting for the vast numbers of SNVs that fail to reach significance [18–20]. For this reason, and because SNV-level heritability is typically scattered throughout the genome [21,22], the expression of many phenotypes is thought to be modified by a polygenic component involving hundreds of causal variants that —collectively—have subtle effects on the activity of a similarly large number of genes [23,24]. Advanced approaches to functional analyses for GWAS data therefore strive to incorporate the

signal contained in practically all SNVs, irrespective of their individual, statistical significance [25].

Gene-set analyses surmise that the collective impact of subtle perturbations to multiple genes operating in a biological process, can have a marked effect on that process, and their goal is to identify such processes [26,27]. Gene-set analyses for GWAS data [28], which were developed to control for confounders like gene size and LD structure, use GWAS results for all tested SNVs to execute two analytical steps. In the first, SNVs are mapped to genes they are thought to affect (Fig 1A), such that SNV-level associations can be used to compute gene scores (Fig 1B). In the second, gene scores are fed into a competitive gene-set analysis, whose comparative nature is designed to identify biological processes that are enriched for phenotype association (Fig 1C). The latter step is crucial, because gene scores are generally modest and negligible in isolation (reflecting SNV-level associations). In the proper biological context, relatively higher-scoring genes may group together to expose the involvement of specific biological processes in the etiology of a phenotype. The most widely used gene-set analysis tools for GWAS data are VEGAS2 [29], Pascal [30], and MAGMA [31].

The mapping of SNVs to genes clearly determines the outcome of gene-set analyses for GWAS data. Normally, SNVs are mapped to genes with which they overlap. To account for

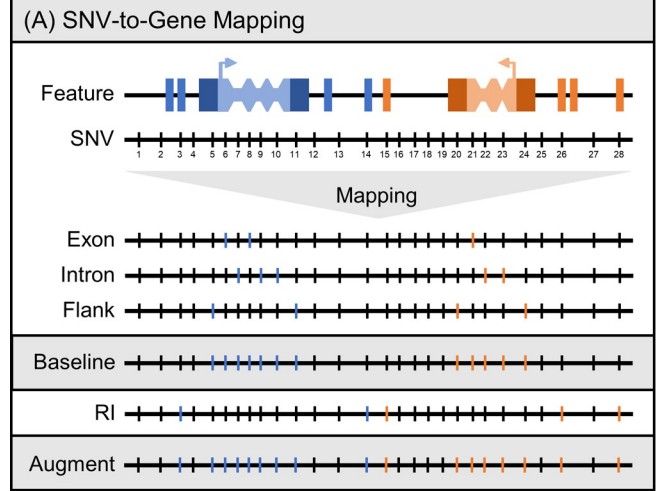

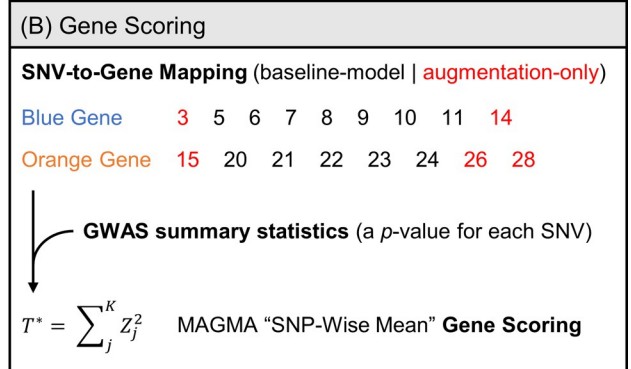

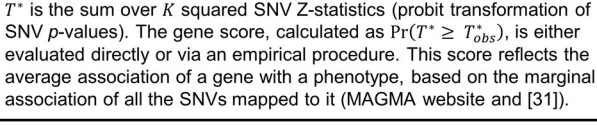

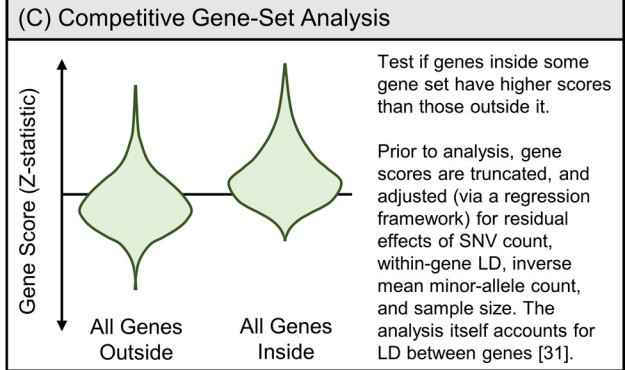

**Fig 1. An overview of gene-set analyses for GWAS data.** Gene-set analyses for GWAS data require that SNVs are mapped to genes (A) SNV-to-Gene Mapping, so that GWAS results (that is, SNV-level associations for all tested SNVs) can be used to test genes for association with a phenotype (B) Gene Scoring. The resulting gene scores are then used to identify biological processes enriched for phenotype association (C) Competitive Gene-Set Analysis. Everything is illustrated in the context of our work, which leveraged MAGMA, one of the most popular tools for gene-set analysis for GWAS data. Abbreviations: RI (regulatory interaction).

regulatory elements, it is common to extend gene bodies beforehand with flanks. Diverse flanks sizes have been used across studies, such as 10kb [31], 20kb [32], 50kb [33], 100kb [34], 200kb [35] and 500kb [36]. Whereas small flanks cannot capture long-range regulatory interactions (RIs), which may span hundreds of kilobases [37,38], large flanks inevitably result in more noisy mappings, effectively diluting true signals. To overcome these limitations, recent studies explored the prospect of improving gene-set analyses for GWAS data by leveraging datasets of RIs to map SNVs to genes. For instance, eMAGMA mapped SNVs to genes through expression quantitative-trait loci (eQTL) data, and then used the gene-scoring framework of MAGMA to identify novel risk genes for major depression [39] as well as other neurological phenotypes [40]. Separately, H-MAGMA, relying also on the MAGMA tool, augmented a more traditional mapping scheme with RIs inferred from chromatin-conformation capture (HiC) of brain tissue, to improve the identification of brain-disorder risk genes [41].

In this study, we used MAGMA to systematically investigate the apparent benefits of leveraging datasets of RIs to map SNVs to genes in the context of gene-set analyses for GWAS data. Specifically, we evaluated how augmenting (that is, building on-top-of) the default, flank-based mapping scheme of MAGMA with datasets of RIs, compared in terms of numbers, and identities, of genes and gene sets detected as statistically significant (Fig 1). We integrated various kinds of datasets of RIs, including computationally-predicted enhancer-promoter mappings (EPMs), as well as experimentally-determined interactions between genomic loci based on HiC and promoter-capture HiC (pc-HiC). Our evaluations were performed on GWAS data for ten different phenotypes. We show that, while integrating datasets of RIs into gene-set analyses for GWAS data can strengthen the association of biologically-relevant gene sets with phenotypes, this approach introduces pitfalls and confounding at several stages. By introducing an array of control strategies to distinguish between genuine results and spurious discoveries, we present a critical approach for integrating datasets of RIs into gene-set analyses for GWAS data.

## Results

We set out to investigate to what extent augmenting the mapping of SNVs to genes with datasets of RIs improves gene-set analyses for GWAS data. To this end, we used MAGMA (v1.08), a popular gene-set analysis tool for GWAS data, which also served as the basis of other studies relevant to our own: eMAGMA [39,40] and H-MAGMA [41]. We designated the default SNV-to-gene mapping scheme of MAGMA as our baseline model (that is, we mapped SNVs to a gene if they overlapped with its gene body or were located within 10kb from it), and we augmented this baseline model with an assortment of RIs (that is, we mapped additional SNVs to a gene if they overlapped with one of its regulatory elements) (Table 1), including: EPM datasets, HiC datasets, promoter-capture HiC (pc-HiC) datasets, and cell-map (cMap) datasets (that is, various independently-published datasets of RIs for specific cell-types). We then analyzed GWAS data (summary statistics) for ten phenotypes related to different traits and diseases (Table 2), and evaluated the effects of augmentation by comparing numbers, and identities, of genes and gene sets detected as statistically significant between SNV-to-gene mappings. Note that the pc-HiC and cMap datasets used for augmentation, differed according to phenotype (S1 Table).

### Gene-scoring analyses

We first investigated the effects of SNV-to-gene mapping augmentations on gene scores of ~18K protein-coding genes. Gene scores were calculated with MAGMA's "SNP-wise mean" method, and then adjusted by MAGMA for confounders (that is, outlying gene scores were

**Table 1. Overview of SNV-to-gene mappings incorporated into our analyses (baseline model in bold font).**

| Shared Features | Flanks^ | Regulatory Interactions Category | Regulatory Interactions Resource | Number of Regulatory Interactions[+] | Number of Genes[+] | Mean Size (bp) of Regulatory Elements[+, #] | Remarks |
|---|---|---|---|---|---|---|---|
| Gene Body | U0D0 | - | - | - | - | - | - |
| Gene Body | U2D0 | - | - | - | - | - | - |
| Gene Body | U2D2 | - | - | - | - | - | - |
| **Gene Body** | **U10D10** | - | - | - | - | - | - |
| Gene Body | U20D20 | - | - | - | - | - | - |
| Gene Body | U35D35 | - | - | - | - | - | - |
| Gene Body | U50D50 | - | - | - | - | - | - |
| Gene Body | U100D100 | - | - | - | - | - | - |
| Gene Body | U250D250 | - | - | - | - | - | - |
| Gene Body | U500D500 | - | - | - | - | - | - |
| Gene Body | U1000D1000 | - | - | - | - | - | - |
| Gene Body | U2D2 | EPM | DHS07 [Local] | 49,650 | 6,664 | 653 | - |
| Gene Body | U2D2 | EPM | FOCS [42] | 118,961 | 14,893 | 399 | ∪(All 4 maps)* |
| Gene Body | U2D2 | EPM | GeneHancer [43] | 191,173 | 17,971 | 1,021 | - |
| Gene Body | U2D2 | EPM | JEME [44] | 255,439 | 17,435 | 927 | - |
| Gene Body | U10D10 | EPM | DHS07 [Local] | 49,650 | 6,664 | 653 | - |
| Gene Body | U10D10 | EPM | FOCS [42] | 118,961 | 14,893 | 399 | ∪(All 4 maps)* |
| Gene Body | U10D10 | EPM | GeneHancer [43] | 191,173 | 17,971 | 1,021 | - |
| Gene Body | U10D10 | EPM | JEME [44] | 255,439 | 17,435 | 927 | - |
| Gene Body | U10D10 | EPM | PsychENCODE [45] | 51,214 | 5,069 | 442 | - |
| Gene Body | U10D10 | HiC | Adult Brain [45] | 33,709 | 11,435 | 13,153 | - |
| Gene Body | U10D10 | HiC | Fetal Brain [46] | 61,535 | 14,041 | 16,097 | - |
| Gene Body | U10D10 | pc-HiC | Selected (See Table A in S1 Table) [47] | | | | - |
| Gene Body | U10D10 | pc-HiC | Global [47] | 424,182 | 17,517 | 5,872 | ∪(All samples)! |
| Gene Body | U10D10 | cMap | Selected (See Table B in S1 Table) [48–52] | | | | - |

^ Flanks: UX (U; upstream from transcription start-site) and DY (Y; downstream from transcription end-site), where X and Y are flank size in kb.

[+] Numbers before annotation with SNVs (that is, including regulatory elements within—or linked to genes overlapping with—the MHC region).

[#] Calculated across all regulatory interactions (that is, the same regulatory element may be considered multiple times if linked to multiple genes).

* Union of datasets (FANTOM5, ENCODE, Roadmap, GRO-seq).

! Union of promoter-other datasets (AD2, AO, BL1, FC, EG2, FT2, GA, GM, H1, HCmerge, IMR90, LG, LI11, LV, ME, MSC, NPC, OV2, PA, PO3, RA3, RV, SB, SG1, SX, TB, TH1).

truncated, before using regression to correct for residual effects of SNV count, within-gene LD, inverse mean minor-allele count, and sample size). Every gene score is a test statistic reflecting the mean association of a gene with a phenotype, based on the association of each SNV mapped to it (see the MAGMA publication for details [31]). Given the self-contained nature of these tests (that is, each gene is tested independently of, and not relative to, all other genes, against a null hypothesis of no association), and that regulatory elements known to be enriched for strong SNV-level associations to diverse phenotypes [4–11], we expected augmentation with RIs to increase the number of genes significantly associated with any phenotype. Thus, we started off by comparing between numbers of significant genes detected (FDR-adjusted $p$-value $< 0.05$) with different SNV-to-gene mappings for each phenotype. As expected, augmentation with RIs nearly always increased the number of significant genes detected for any phenotype (Table A in S2 Table). However, the magnitude of increase varied

**Table 2. GWAS datasets (summary statistics) analyzed.**

| Phenotype[*] | Reference | Number of SNVs[^] | Number of Genes[+] | Number of Gene-Sets[+] |
|---|---|---|---|---|
| Alzheimer's Disease | [53] | 7,035,668 | 18,461 | 2,647 |
| Atrial Fibrillation | [54] | 7,275,249 | 19,257 | 2,647 |
| Bone Density | [55] | 6,269,624 | 18,283 | 2,647 |
| Breast Cancer | [56] | 7,377,348 | 19,325 | 2,647 |
| C-Artery Disease | [57] | 6,518,657 | 18,241 | 2,647 |
| Crohn's Disease | [58] | 6,412,306 | 18,282 | 2,647 |
| Mac. Degeneration | [59] | 6,626,920 | 18,881 | 2,647 |
| Prostate Cancer | [60] | 7,386,742 | 19,328 | 2,647 |
| Schizophrenia | [61] | 6,307,153 | 18,555 | 2,647 |
| Type-2 Diabetes | [62] | 6,486,861 | 18,359 | 2,647 |

[*] Phenotype abbreviations: C-Artery Disease (coronary-artery disease) and Mac. Degeneration (Macular Degeneration).

[^] Numbers refer to those in the processed GWAS summary-statistics datasets (see Methods for details).

[+] For each phenotype, we focused our analyses exclusively on genes or gene sets with a score for every SNV-to-gene mapping (see Methods).

considerably, and appeared to depend more on the dataset of RIs used for augmentation, than on its presumed relevance to a phenotype. For instance, augmentation with the fetal-brain HiC dataset of RIs often increased the number of significant genes more than that with other datasets of RIs, even for non-neurological phenotypes.

For comparison, we then likewise evaluated the performance of flank-based mappings. We reasoned that large flanks would capture many genuine RIs, albeit at the expense of added noise. Notably, for all phenotypes we detected more-and-more significant genes with increasingly large flanks (Table B in S2 Table). In turn, this led us to suspect that the number of significant genes detected was positively correlated with the "coverage" of a mapping (calculated as the number of non-redundant base pairs covered by all features of a gene—the gene body, flanks, and regulatory elements—summed across all genes). Hence, for each phenotype we plotted the number of significant genes detected against the coverage for diverse mappings, which confirmed our suspicion (Fig 2A; note the large coverage associated with the fetal-brain HiC dataset of RIs). To understand this trend, we studied the responses of individual genes (as opposed to genes overall) to augmentation, and established that the main effect of augmentation was to make new genes significant (that is, in most cases, comparatively few genes lost significance) (Fig 2B and S3 Table). In essence, larger-and-larger augmentations progressively expanded the baseline-model list of significant genes. We reproduced these findings with MAGMA's unadjusted gene scores (that is, gene scores prior to correction for residual effects), as well as with gene scores calculated by Pascal [30] (a distinct gene-set analysis tool for GWAS data) (S4 Table).

Given the consistent trend with coverage, we became concerned that augmentations were confounding gene scores in a way that any mere increase in the size of genes (as opposed to an increase linked specifically to augmentation with genuine flanks and RIs) would yield more significant genes. As a matter of fact, we realized that augmentation was not just affecting the size of genes, but also, for instance, their internal LD structure. Hence, to address concerns over confounding in general (that is, without assuming its sources), we developed a permutation control, called extragenic p-value push (EPVP), which operates on GWAS summary statistics, and randomly assigns background SNV-level associations to "extragenic SNVs" of individual genes (that is, SNVs mapped to a gene via augmentation, but not via the baseline

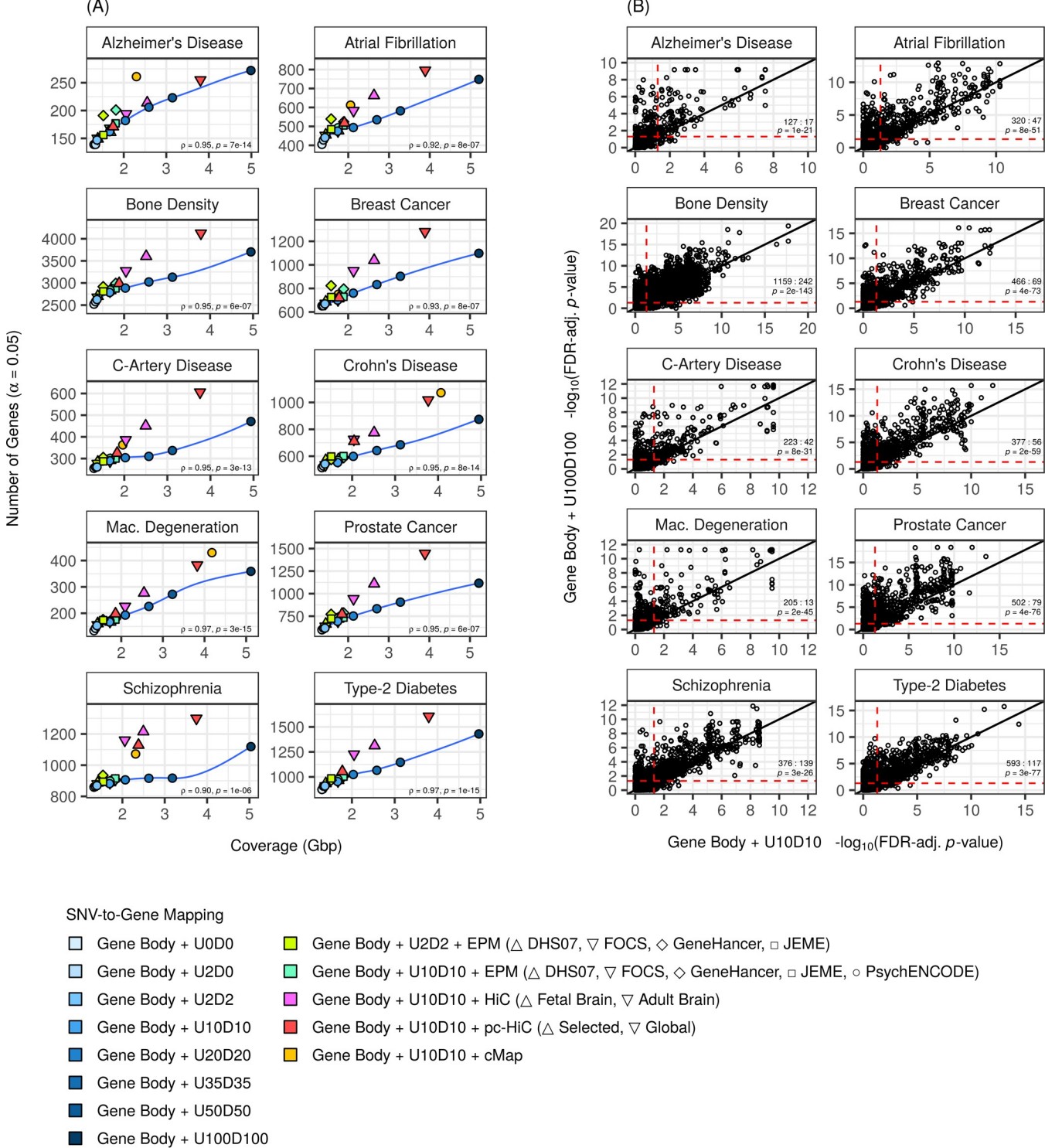

**Fig 2. Larger SNV-to-gene mappings (by coverage) yield more significant genes. (A)** The number of significant genes detected for diverse SNV-to-gene mappings as a function of their coverage (that is, the number of non-redundant base pairs covered by all features–gene bodies, flanks, and regulatory elements —defining a gene, summed across all genes). Blue, solid line shows trend (loess smooth) for flank-based mappings (including Gene Body + U0D0). For each phenotype, we reported Spearman's rank correlation coefficient (ρ) and its associated *p*-value (*p*) for a test of positive correlation based on all mappings (including gene bodies with 250kb flanks, gene bodies with 500kb flanks, and gene bodies with 1000kb flanks, which are not depicted on the graphs themselves but provided in Table B in S2 Table). **(B)** Gene scores of individual genes (circles), comparing between a mapping with large flanks (that is, 100kb flanks) and a mapping with small flanks (that is, 10kb flanks, which is the baseline model). Black, solid line shows the identity line. Red, dashed line shows the significance

cut-off ($\alpha = 0.05$). We tested (binomial test) if there were more novel genes (N; genes significant only with large flanks) than lost genes (L; genes significant only with small flanks), against the null hypothesis that both outcomes are equally likely or that losing is more likely. Counts (N:L) and the FDR-adjusted $p$-value ($p$) of each test (that is, FDR-adjusted within each phenotype across all the mappings reported in S3 Table) are reported. **(A) and (B)** Flanks were defined as regions extending from gene bodies; specifically, as UX (U; upstream from the transcription start-site) and DY (Y; downstream from the transcription end-site), where X and Y are flank size in kb. Phenotype abbreviations: C-Artery Disease (coronary-artery disease); Mac. Degeneration (macular degeneration).

model), while preserving the actual mapping of SNVs to genes (S1 Fig and Methods). In other words, EPVP yields gene scores under conditions analogous to using matched, random RIs or flanks, in place of the genuine ones, but with limited disruption to potential confounders (since, the number and identities of SNVs mapped to each gene remain unchanged). We used this analogy (that is "random, matched RIs") throughout our manuscript.

We executed EPVP for seven augmented mappings for each phenotype (using 20 independent permutations in each case). We observed (1) that genuine augmentations always yielded more significant genes than matched, random augmentations, and (2) that the number of significant genes detected with the baseline model, was usually either greater than, or similar to, that detected with random augmentations of the baseline model (Fig 3 and S5 Table). This meant that the observed trend with coverage was not merely explained by confounders. Indeed, it reassured us that SNV-level associations (that is, the GWAS $p$-values permuted by EPVP) were a key driver of this trend and of gene scores in general. Moreover, the first observation indicated that SNVs located within genuine flanks and regulatory elements were generally enriched for strong phenotype associations. Still, EPVP was unable to entirely rule-out confounding effects from augmentation. In fact, large (by coverage), random augmentations, intermittently increased the number of significant genes relative to the baseline model (for example, see results with random RIs matched for the global pc-HiC dataset of RIs in Fig 3) and exposed an apparent effect of gene size on gene scores (S2 and S3 Figs). While it was beyond the scope of our work to study this effect in more depth, we favored its conservative implication: to always evaluate the effects of a genuine augmentation against matched, random augmentation.

## Gene-set analyses

Next, we examined the effects of SNV-to-gene mapping augmentations on gene-set scores. We fed the previously-calculated gene scores into MAGMA's competitive gene-set analyses, to test 2,647 GO biological-process gene sets [63] for enrichment for phenotype association. Unlike gene scores, these analyses are comparative, because, for each gene set (a test set), the phenotype association amongst its genes (reflected by the gene scores) is tested relative to that amongst all other genes. We first compared the number of significant gene sets detected between mappings for each phenotype (FDR-adjusted $p$-value $< 0.05$). As GO gene sets can be highly overlapping [64], we used the "rrvgo" package in R [65] to define, per combination of mapping and phenotype, a non-redundant subset of significant gene sets on which we focused our evaluations. Most augmentations of the baseline model did not yield a marked change in the number of significant gene sets detected (S6 Table), and, to be more precise, the significant gene sets themselves were often shared between the baseline model and some augmented model (S7 Table). That being said, augmentations that greatly increased the number of significant genes detected, such as 100kb flanks, attenuated gene-set scores relative to the baseline model and were recurrently associated with fewer significant gene sets (Fig 4 and S6 Table).

We then set out to identify the specific gene sets that gained from augmentation of the baseline model (that is, demonstrated a stronger enrichment for phenotype association with, than without, augmentation). First, focusing on 562 significant gene sets identified for selected

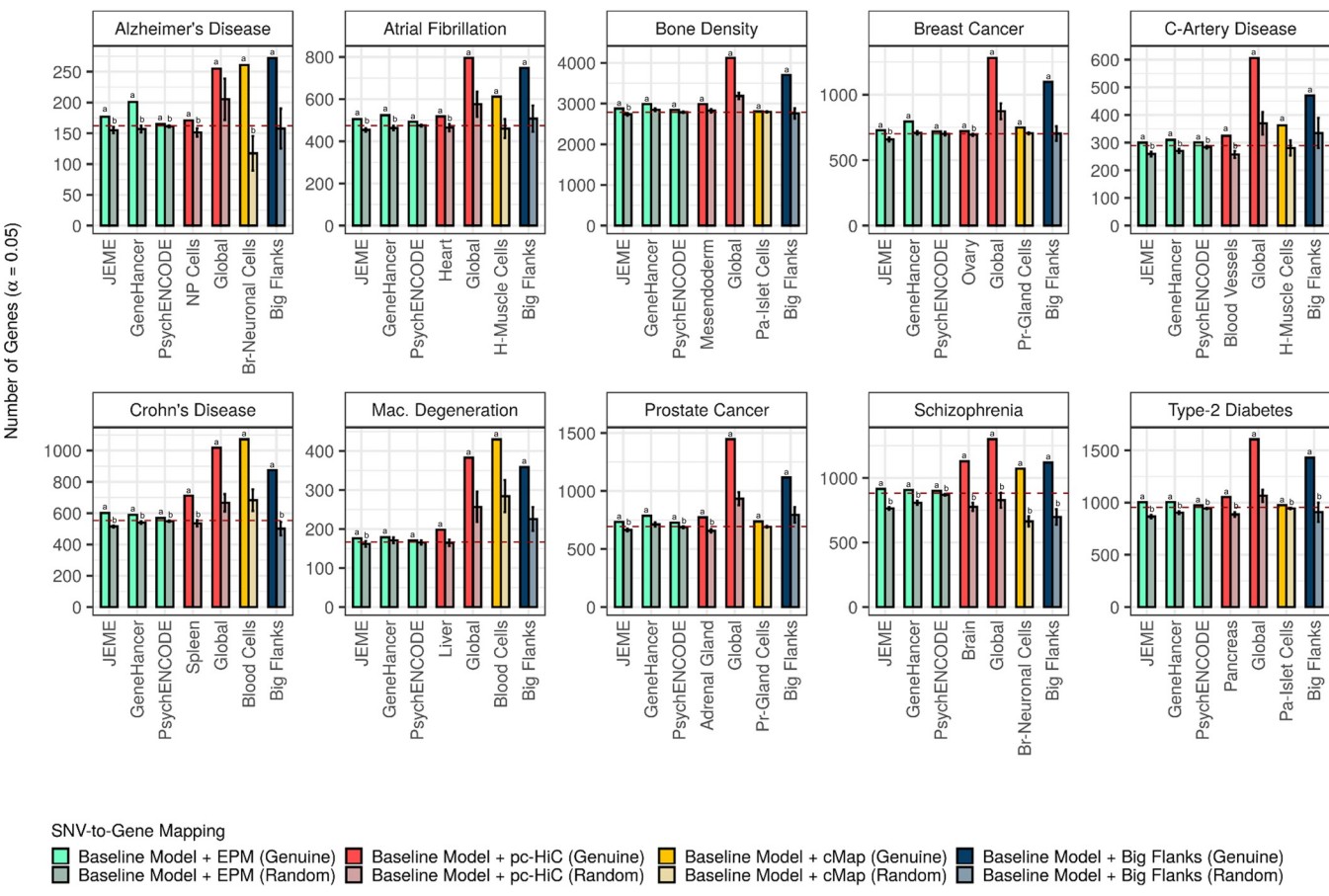

**Fig 3. Genuine augmentations are associated with more significant genes than matched, random augmentations.** The number of significant genes detected for selected, genuine augmentations (saturated bars), in comparison to matched, random augmentations (desaturated bars). For random augmentations, counts and error bars represent the mean and standard deviation, respectively (based on 20 independent permutations of EPVP). Red, dashed line shows the number of significant genes detected with the baseline model itself. We tested (T-test) if there were more significant genes with genuine augmentation than with matched, random augmentation ([a] above a saturated bar indicates that the $p$-value of the test, which was FDR-adjusted across all mappings within each phenotype, was smaller than 0.05). We then tested, in a similar way, if there were more significant genes with the baseline model than with random augmentation of the baseline model ([b] above a desaturated bar for an FDR-adjusted $p$-value smaller than 0.05). The augmentation, "Big Flanks", refers to gene bodies with 100kb upstream- and downstream flanks (note, as part of the baseline model, the first 10kb on either side of a gene body were not permuted by EPVP). Phenotype abbreviations: C-Artery Disease (coronary-artery disease); Mac. Degeneration (macular degeneration). Mapping abbreviations: Br-Neuronal Cells (brain-neuronal cells); H-Muscle Cells (heart-muscle cells); NP Cells (neural-progenitor cells); Pa-Islet Cells (pancreatic-islet cells); Pr-Gland Cells (prostate-gland cells).

combinations of phenotypes and augmented mappings, only 308 (308/562) gained from augmentation (Fig 5 and S8 Table). Then, to scrutinize the possibility that a gain was non-specific to genuine augmentation, we applied our previously described permutation control, EPVP, to obtain gene-set scores under conditions as if genuine augmentations had been replaced with random counterparts (S1 Fig and Methods). For each of the 308 gene sets that gained from augmentation, we compared its score obtained with genuine augmentation, to its scores obtained with matched, random augmentations (based on 20 independent permutations of EPVP). In this way, we assigned each gene set to one of three validation categories: (a) strongly validated (a genuine score at least two SD above the mean, random score), (b) mildly validated (a genuine score at least one, but less than two, SD above the mean, random score), and (c) invalidated (the rest). Remarkably, over a third of all gains from augmentation (119/308) were invalidated by the EPVP procedure, which means that these gene sets gained no more from

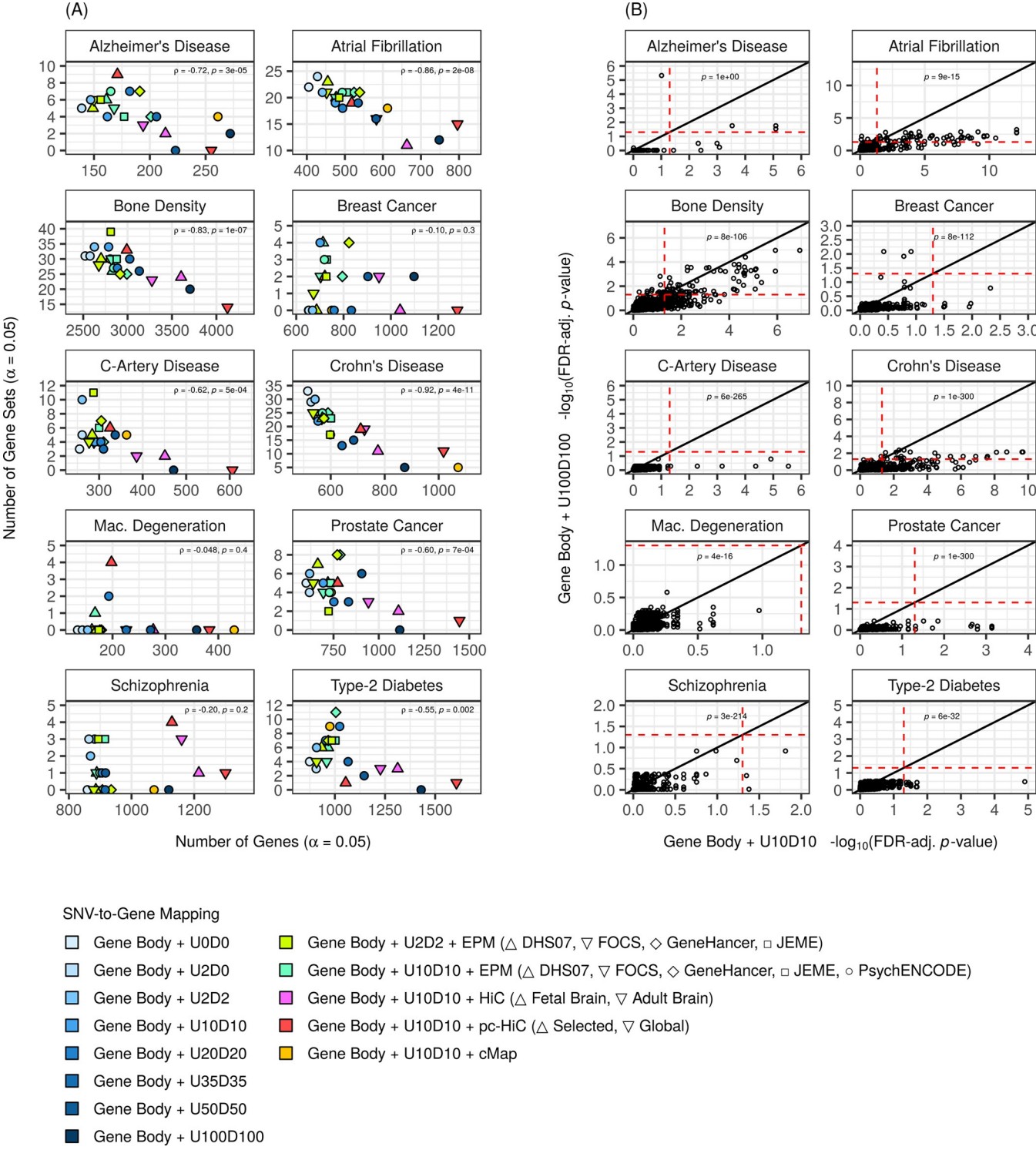

**Fig 4. Mappings that yield more significant genes are associated with fewer significant gene sets. (A)** The number of non-redundant, significant gene sets detected as a function of the number of significant genes detected for diverse mappings. For each phenotype, we reported Spearman's rank correlation coefficient ($\rho$) and its associated $p$-value ($p$) for a test of negative correlation based on all mappings (including gene bodies with 250kb flanks, gene bodies with 500kb flanks, and gene bodies with 1000kb flanks, which are not depicted on the graphs themselves but provided in Table B in S6 Table). **(B)** Gene-set scores of individual gene sets (circles), comparing between a mapping with large flanks (that is, 100kb flanks) and a mapping with small flanks (that is, 10kb flanks, which is the baseline model). All gene sets are shown (irrespective of redundancy). Black, solid line shows the identity line. Red, dashed line shows the significance cut-off ($\alpha = 0.05$). We tested (one-sided, paired Wilcoxon test) if there was a tendency for gene-set scores to be attenuated with large flanks

(relative to small flanks). The FDR-adjusted *p*-value (*p*) of each test (that is, FDR-adjusted within each phenotype across all the mappings reported in S6 Table) is shown (note that extreme *p*-values were truncated to 1e-300 for readability). **(A) and (B)** Flanks were defined as regions extending from gene bodies; specifically, as UX (U; upstream from the transcription start-site) and DY (Y; downstream from the transcription end-site), where X and Y are flank size in kb. Phenotype abbreviations: C-Artery Disease (coronary-artery disease); Mac. Degeneration (macular degeneration).

the genuine augmentation than from matched, random augmentation (Figs 6 and S4, and S9 Table).

We next considered that, based on the way a gain was defined, some of the 189 validated (87 mildly and 102 strongly) gains that we had observed for gene sets, could be trivial (that is, driven by very few genes). Rather than focus on the magnitude of a gain (that is, the gene-set score difference between an augmented model and the baseline model) directly, we developed a simple strategy, which we called iterative reduction (IRED), to distinguish between robust gains (that is, gains driven by numerous genes in a gene set improving from augmentation) and non-robust gains (that is, gains driven by just a few such genes). IRED ranks genes from a selected gene set according to their magnitude of gain (defined in the same way as for a gene set, but using gene scores), before cumulatively removing one gene after another from the selected gene set, starting with the gene that gained the most from augmentation. At each iteration, gene-set analysis is executed again for both the augmented model and the baseline model. The analytical end-point is the cumulative number of genes that must be removed for the gain at the level of the gene set to be lost.

We executed IRED for nearly all gene sets with validated gains (187, since two gene sets contained too few genes to execute the procedure). The cumulative number of genes that had to be removed from a gene set for its gain to be lost, was frequently very small (that is, only one gene for 36 gene sets, two genes for 18 gene sets, and three genes for 23 gene sets), which would limit further functional analyses. For the remaining 110 gene sets with validated gains, at least the top four genes (by their magnitude of gain) had to be removed before the gain at the level of the gene set itself was lost (that is, robust gains). To illustrate the concept of robustness of a gain, we compared how the difference in gene-set score between an augmented model and the baseline model at each iteration, was related to the difference in the gene score of the top-gaining gene present at an iteration. We present an example of a robust gain (Fig 7A) and a non-robust gain (Fig 7B). We summarized the numbers of non-robust and robust, validated gains amongst gene sets examined for the selected combinations of phenotypes and mappings (S10 Table), and for reference, we listed the IRED results for all 187 gene sets evaluated (S11 Table).

### Pinpointing genes and regulatory elements that carry the gain of a gene set from augmentation

For our final analyses, we focused on four selected gene sets that demonstrated robust, validated gains from augmentation of the baseline model with RIs. For each of these case studies we (i) illustrated the validation of the gain with EPVP, (ii) used IRED to identify a group of top-gaining genes that were collectively relevant to the gain, and (iii) used a qualitative approach to pinpoint the regulatory elements of those genes that carried the gain. The aim of these case studies was to show how augmenting the mapping of SNVs to genes with datasets of RIs, can help link regulatory elements to phenotypes through the genes and biological processes they regulate.

### Schizophrenia and RIs from the pc-HiC dataset of brain: Post-synaptic chemical transmission

Augmentation of the baseline model with RIs from the pc-HiC dataset of brain, led to the detection of four gene sets significantly associated with schizophrenia. All four gene sets gained

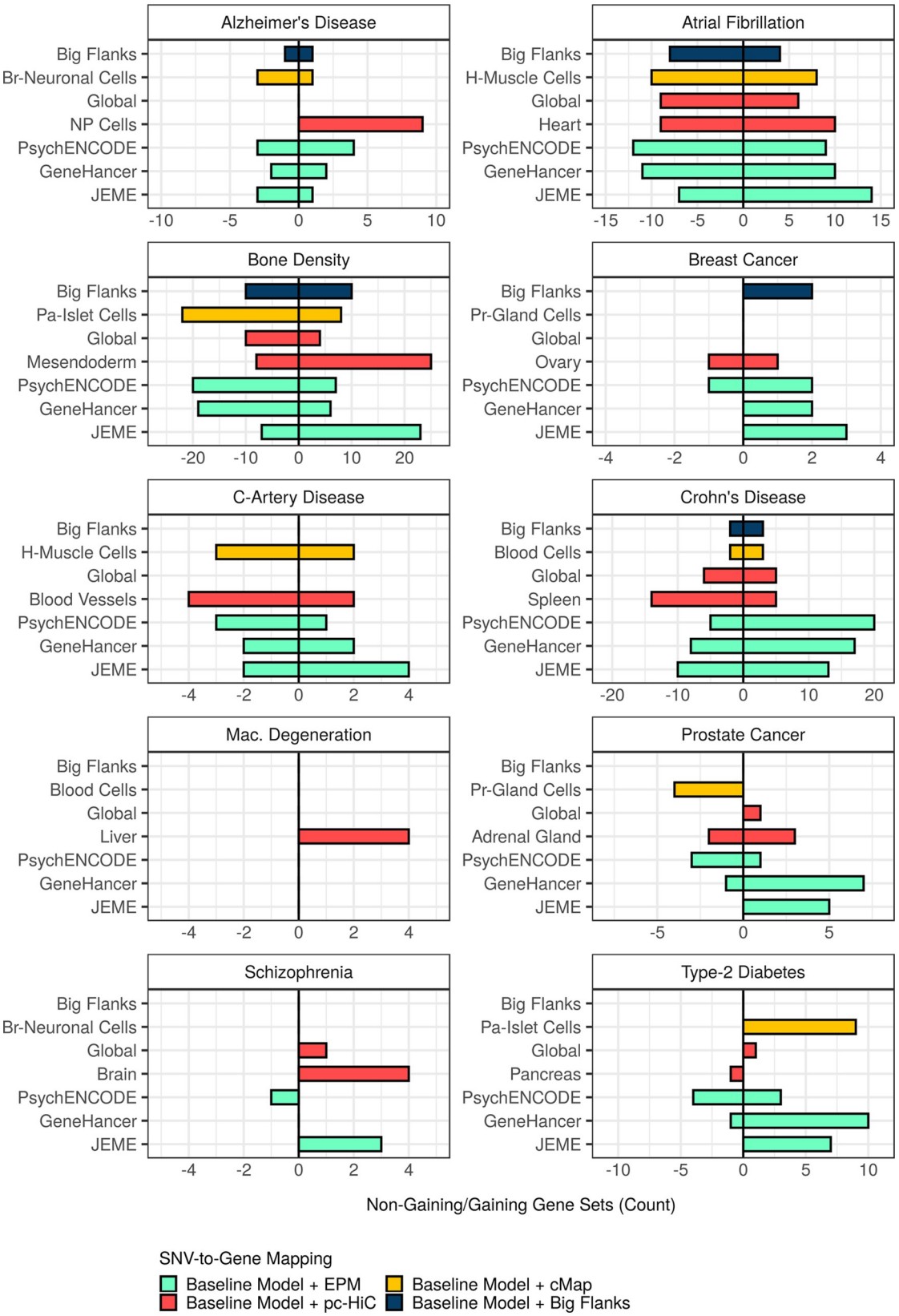

**Fig 5. Gene sets detected as significant with some augmentation, do not necessarily gain from that augmentation.** Significant gene sets detected for selected augmentations for each phenotype were stratified according to whether they gained from augmentation (that is, demonstrated a stronger enrichment for phenotype association with, than without, augmentation) or not. A bar directed to the

right-hand side (positive values) denotes the number of gene sets that gained ("gaining"), and a bar directed to the left-hand side (negative values) denotes the number of gene sets that did not gain ("non-gaining"). The augmentation, "Big Flanks", refers to gene bodies with 100kb upstream- and downstream flanks. Phenotype abbreviations: C-Artery Disease (coronary-artery disease); Mac. Degeneration (macular degeneration). Mapping abbreviations: Br-Neuronal Cells (brain-neuronal cells); H-Muscle Cells (heart-muscle cells); NP Cells (neural-progenitor cells); Pa-Islet Cells (pancreatic-islet cells); Pr-Gland Cells (prostate-gland cells).

from augmentation and all gains were validated by EPVP (two strongly and two mildly) (S5A Fig). Using IRED, we found that only two of these gains were robust: the strongly-validated gain for the *post-synaptic chemical transmission* gene set and the mildly-validated gain for the *adult behavior* gene set (S11 Table). Focusing on the highly-relevant *post-synaptic chemical transmission* gene set [66,67], which contained 109 genes in total (note that, 107 of these genes had a score with the baseline model, one additional gene had a score exclusively with augmentation, and one gene did not have a score with either model), we found that six top-gaining genes (namely and in order, *CHRNB2*, *GHRL*, *AKT1*, *CHRNE*, *SLC17A7*, and *P2RX1*) had to be cumulatively removed from this gene set for its gain to be lost (S5B Fig).

We then compared the gene scores of the ten top-gaining genes from this gene set between the baseline model, the baseline model augmented with genuine RIs, as well as the baseline model augmented with matched, random RIs (based on 20 independent permutations of EPVP) (Fig 8A, right panel). We accompanied each score comparison with an illustration of the relevant gene in its wider genomic context, in order to relate gains to regulatory elements (Fig 8A, left panel). In this way, we linked the gain for *CHRNB2*, a gene encoding an acetylcholine receptor subunit, to a cluster of regulatory elements situated in a schizophrenia-associated locus ~330kb downstream of the transcription-start site of the gene. In a similar fashion but involving even longer-range RIs (~700-1000kb), we implicated a second gene encoding an acetylcholine receptor subunit, *CHRNE*, in schizophrenia. Notably, our illustration shows that neither *CHRNB2* nor *CHRNE* contained any remarkable schizophrenia-associated SNVs within their vicinity (that is, the baseline model), and we implicated these two genes in schizophrenia exclusively through the aforementioned long-range RIs. Acetylcholine receptors are believed to be highly relevant to cognitive deficits in schizophrenia, although most research to date has been dedicated to a different acetylcholine receptor known as *CHRNA7* [68–70].

## Type-2 diabetes and RIs from the cMap dataset of pancreatic-islet cells: endocrine system development

Augmentation of the baseline model with RIs based on the cMap dataset of pancreatic-islet cells, led to the detection of nine gene sets significantly associated with type-2 diabetes (S6A Fig). Although all nine gene sets gained from augmentation, only the gain for the *endocrine system development* gene set was strongly validated by the EPVP procedure (that is, four gene sets were mildly validated, and four gene sets were invalidated). The gain for the *endocrine system development* gene set, which contained a total of 127 genes (note that, 125 of these genes had a score with both models, and that the other two genes did not have a score with either model), was lost only after removing the seven top-gaining genes from the gene set (namely and in order, *SOX4*, *CDKN1C*, *MNX1*, *INSM1*, *FOXA2*, *ISL1*, and *CITED2*) (S6B Fig and S11 Table).

Illustrating the ten top-gaining genes, as we did for the previous case study, revealed interesting RIs such as for *CDKN1C* (regulatory elements ~100kb, as well as ~200kb, downstream from the transcription-start site), *SOX4* (regulatory elements between ~600-1000kb upstream from the transcription start-site), *MNX1* (regulatory elements ~150kb upstream from the transcription start-site), and *ISL1* (regulatory elements ~1000kb downstream from the transcription-start site) (Fig 8B). The cluster of regulatory elements for *ISL1* did not overlap with

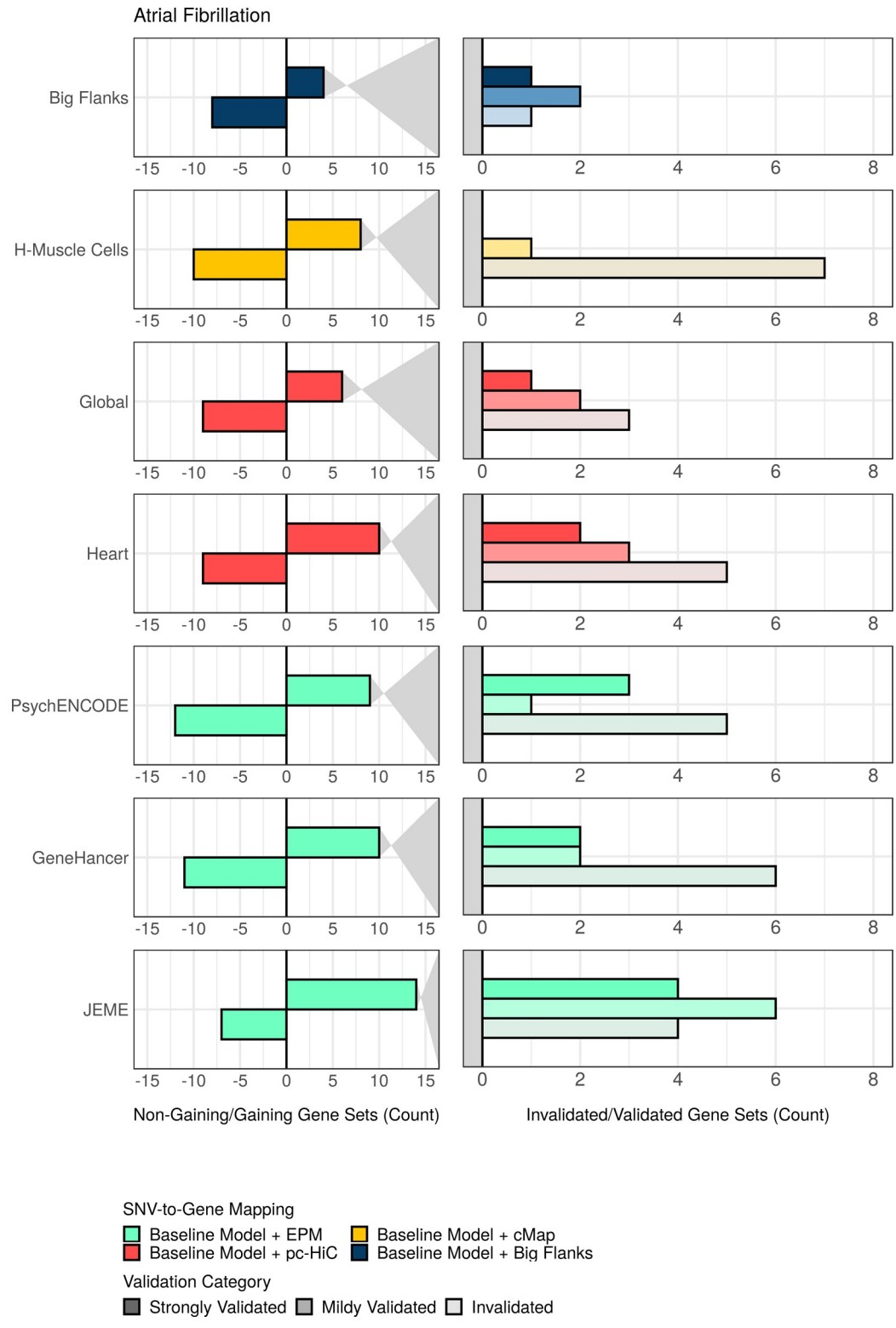

**Fig 6. Significant gene sets that gain from augmentation, often gain no more from genuine augmentation than from matched, random augmentation.** Left panel: the numbers of gaining (bars directed rightwards and positive values) and non-gaining (bars directed leftwards and negative values) gene sets amongst significant gene sets detected for atrial

fibrillation with selected augmentations (note, this is similar to the graph for atrial fibrillation in Fig 5). Right panel: significant gene sets that gained from augmentation were stratified and counted according to three validation categories (strongly validated, mildly validated, and invalidated), which defined how pronounced a gain was with genuine augmentation, over that with matched, random augmentation. The augmentation, "Big Flanks", refers to gene bodies with 100kb upstream- and downstream flanks (note, as part of the baseline model, the first 10kb on either side of a gene body were not permuted by EPVP). Mapping abbreviations: H-Muscle Cells (heart-muscle cells). Refer to S4 Fig for similar graphs for the other phenotypes.

protein-coding genes, and was independently associated with *ISL1* gene expression (GTEx eQTL data [71]), thus endorsing it as an interesting locus for future research. *ISL1* is a transcription factor required for the development of pancreatic-cell lineages [72–74]. Separately, although the regulatory elements implicating *SOX4* –a gene involved in insulin secretion– overlapped with a different gene (*CDKAL1*), this distal locus has indeed previously been studied and found to incriminate *SOX4* in the context of type-2 diabetes [75,76].

Two additional case studies are discussed in supplementary material: *Bone density and RIs from the global pc-HiC dataset*: *response to BMP* (S7 Fig) and *Prostate cancer and RIs from the EPM dataset of GeneHancer*: *regulation of embryonic development* (S8 Fig).

## Discussion and conclusions

Gene-set analyses for GWAS data consider the signal contained in practically all genotyped and imputed SNVs to detect functionally-annotated collections of genes that are enriched for

(A)  Atrial Fibrillation
Actin-Mediated Cell Contraction (EPM: JEME)

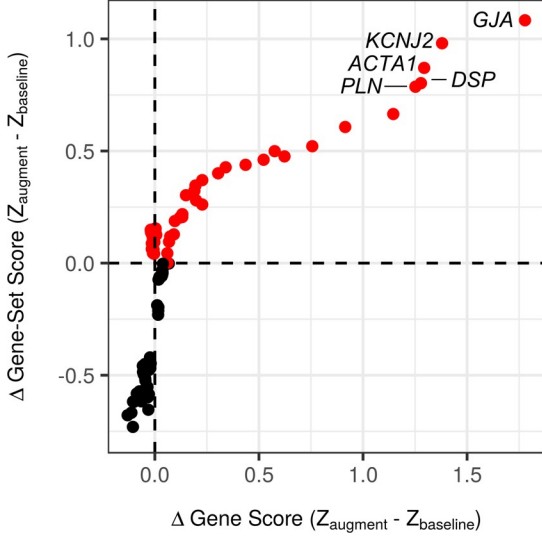

(B)  Crohn's Disease
Positive T-Cell Selection (EPM: PsychENCODE)

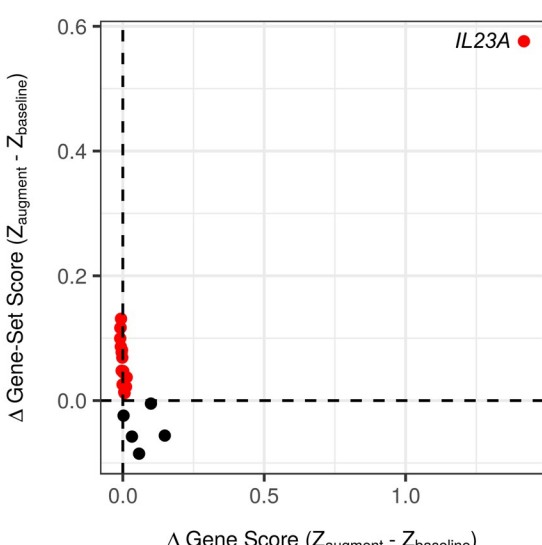

**Fig 7. The IRED procedure for identifying robust gains from augmentation for gene sets.** Genes were iteratively and cumulatively removed, one-by-one, from a selected gene set (beginning with the top-gaining gene, then the second-biggest gaining gene, and so forth). At each iteration, the impact on the gain of the gene set itself was inspected. One dot is shown for each iteration. Each dot represents the gene-set score difference (determined following probit transformations of one minus each FDR-adjusted, upper-tail *p*-value) in relation to the difference in the gene score (as used in gene-set analysis and for ranking the genes for IRED, before multiple-testing correction) of the top-gaining gene still present at an iteration. Given the nature of the procedure, progressive iterations are traced by moving from right to left on a graph. For clarity, red dots mean that the gene set demonstrates a gain from augmentation (otherwise, dots are black). The number of iterations for a gain to be lost for the first time (that is, the number of red dots before the first black dot is encountered when moving from right to left), is counted to assess robustness of a gain of a gene set. For both graphs, some top-gaining genes have been labelled for reference. **(A)** A robust gain (*actin-mediated cell contraction* gene set detected for atrial fibrillation with augmentation from the EPM of JEME dataset of regulatory interactions). **(B)** A non-robust gain (*positive T-cell selection* gene set detected for Crohn's disease with augmentation from the EPM of PsychENCODE dataset of regulatory interactions).

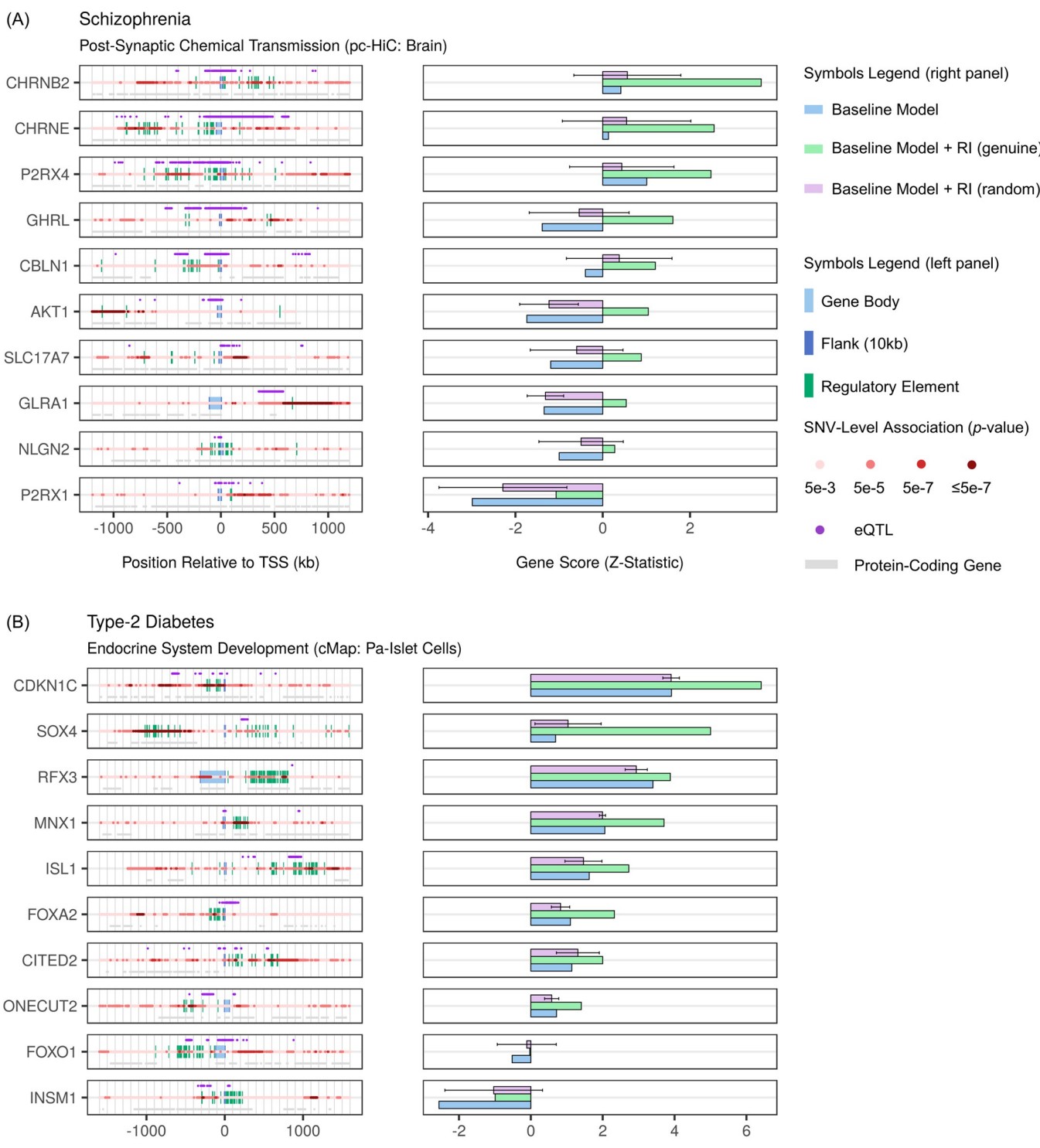

**Fig 8. Identifying genes and regulatory elements that carry gains from particular augmentations for selected gene sets. (A)** Ten top-gaining genes from the schizophrenia-associated gene set, *post-synaptic chemical transmission* (officially, *go_chemical_synaptic_transmission_postsynaptic*), which gained from augmentation of the baseline model with the pc-HiC of brain dataset of regulatory interactions, were inspected to relate regulatory elements to schizophrenia risk. Left panel: illustrating genes in their genomic context to relate their regulatory elements to their gains (right panel), and in turn, to the gain at the level of the gene set itself. Regulatory elements that overlap with strong SNV-level associations for a phenotype (note, SNV-level associations from the GWAS dataset are depicted on the red midline) are of particular interest (note that the legend specifies the most significant SNV-level association represented by a given shade

of red). Regulatory interactions involving interesting regulatory elements may be independently supported by eQTL data (GTEx version 8 for the European population, covering all tissues and cell types) for the same gene (purple dots). Right panel: gene scores (as used in gene-set analysis, before multiple-testing correction) for selected mappings. Bigger, positive scores imply a stronger phenotype association. For random augmentation, gene scores and error bars represent the mean and standard deviation, respectively (based on 20 independent permutations of EPVP). **(B)** Ten top-gaining genes from the type-2 diabetes associated gene-set, *endocrine system development* (officially, *go_endocrine_system_development*), which gained from augmentation with the cMap of Pa-Islet Cells dataset of regulatory interactions, were likewise inspected. Mapping abbreviations: Pa-Islet Cells (pancreatic-islet cells).

phenotype association. To do this, SNVs are mapped to genes they might regulate, before using SNV-level associations to calculate gene scores (that is, each gene is tested independently for association with a phenotype). Gene scores are then used to identify gene sets that are enriched for phenotype association (that is, the phenotype association amongst genes from a gene set is compared to that amongst all other genes). A decade-long effort to improve these analyses from statistical and computational perspectives [29–31,36,77], culminated in the release of MAGMA seven years ago. MAGMA remains one of the most widely-used tools of its kind.

The mapping of SNVs to genes necessarily has a major impact on the outcome of gene-set analyses for GWAS data. Typically, SNVs are mapped to genes in their proximity. However, most causal SNVs are thought to confer their effect via regulatory elements [4–11], which can be hundreds of kilobases away from their target genes [37,38]. This has inspired fresh attempts to improve gene-set analyses for GWAS data; more specifically, by harnessing datasets of RIs to map SNVs to genes. For example, eMAGMA mapped SNVs to genes exclusively through eQTL data, and then applied the gene-scoring framework of MAGMA to identify risk genes for major depression [39] and, more recently, other brain-related phenotypes [40]. Separately, H-MAGMA, relying also on the MAGMA tool, used RIs inferred from HiC of brain tissue to map SNVs to genes, and then performed analyses on GWAS data for several brain-related phenotypes, including schizophrenia and Alzheimer's disease [41]. In essence, the authors of both eMAGMA and H-MAGMA, concluded that using RIs to map SNVs to genes, enables the discovery of new phenotype-modifying genes (compared to more-traditional mapping schemes).

Here, we used MAGMA to systematically investigate the apparent benefits of incorporating RIs into gene-set analyses for GWAS data. We focused on addressing four limitations of eMAGMA and H-MAGMA: (i) the lack of diversity in datasets of RIs and phenotypes analyzed, (ii) the limited use of MAGMA's competitive gene-set analysis, (iii) the omission of a baseline SNV-to-gene mapping scheme (that is, a baseline model) for evaluation purposes, and (iv) a shortage of control strategies to scrutinize results. To this end, we designated MAGMA's default, flank-based mapping scheme as our baseline model, and we augmented this baseline model with a large assortment of RIs. We then analyzed GWAS data for ten different phenotypes. We evaluated the effects of augmentation by comparing numbers, and identities, of genes and gene sets detected as statistically significant between mappings. We explicitly used a baseline model as an evaluation strategy: to ensure that any differences in results between an augmented model and the baseline model were entirely attributable to the augmented component. In addition, having a baseline model allowed us to develop EPVP, a permutation control for comparing genuine augmentation with matched, random augmentation, so as to limit spurious discoveries.

In the first part of our work, we established that the overall effect of augmentation was to expand the baseline-model list of genes significantly associated with a phenotype. We showed that the related increase in the number of significant genes detected, was mostly specific to genuine augmentation (that is, no comparable increase was ever observed with matched, random augmentation). This was consistent with the well-documented enrichment for strong SNV-level associations to phenotypes within promoters and regulatory elements [4–11]. We

then revealed that most augmentations had no marked effect on the number and identities of gene sets significantly enriched for phenotype association. However, augmentations that produced a big increase in the number of significant genes, were typically associated with smaller numbers of significant gene sets. This means that such augmentations improved gene scores broadly (that is, involving numerous genes from many unrelated gene sets), such that no particular gene set(s) "stood out" (in the context of competitive tests for enrichment for phenotype association).

Our results lead to speculation about the meaning of genes detected as statistically significant with augmentation: are most genes that become significant with augmentation truly associated with a phenotype, or do many of these cases represent false positives that result from mapping SNVs with strong phenotype associations to the wrong genes (that is, mismapping)? On one hand, the involvement of very large numbers of genes in the expression of a phenotype is consistent with an omnigenic model [23]. On the other, a failure to detect any gene sets that are significantly enriched for phenotype association (as was common for augmentations that greatly increased the number of significant genes detected) is in-line with the idea of mismapping. Additional concern about mismapping is raised by the fact that the magnitude of increase in the number of significant genes detected with augmentation, seems to depend more on the contribution of an augmentation to coverage, than on its presumed relevance to a phenotype (based on the tissue or cell type from which the RIs were inferred). We point out that our permutation control, EPVP, does not test for mismapping. Mismapping states that genuine augmentations can assign SNVs with strong phenotype associations to the wrong genes (that is, wrong in the context of a phenotype). This occurs when: (i) a regulatory element with no relevance to a phenotype contains a SNV with a strong phenotype association (LD contamination) [11], or (ii) a SNV with a strong phenotype association is located within a relevant regulatory element that interacts with multiple genes, but the effect is mediated via just one of them (that is, it is ambiguous which gene) [10].

In the second part of our work, we focused on gene sets significantly enriched for phenotype association with selected augmentations, and amongst them, we proceeded to identify those that gained from augmentation of the baseline model. Remarkably, we discovered that gains were often non-specific to genuine augmentation. Furthermore, gains that were, still had to be scrutinized for a lack of robustness; often, such gains were eliminated after removing just a few top-gaining genes from the gene set itself. We then focused on gene sets that gained both specifically and robustly from genuine augmentation. From these gene sets, we selected several examples with clear biological relevance to the phenotype with which they were associated. We used these examples to demonstrate an approach for identifying the specific genes, as well as their regulatory elements, that carried the gain of the gene set from augmentation. In this way, our analyses can generate hypotheses regarding the roles of particular regulatory elements (that is, via the genes and biological processes they regulate) in the expression of a phenotype.

Several limitations of our work should be acknowledged. Firstly, we built our evaluations around a single baseline-model scheme for mapping SNVs to genes (that is, gene bodies with 10kb upstream- and downstream flanks). The choice of this particular scheme as the baseline model is debatable, but can be justified on two accounts: (i) both intronic [78] and exonic [79,80] SNVs are known to regulate genes in which they are located and (ii) small flanks can assign proximal regulatory SNVs to genes [81] that may be missed by datasets of RIs [41]. Secondly, our permutation control, EPVP, has a few shortcomings. While EPVP shifts SNV-level associations circularly around chromosomes, in reality, LD blocks are porous and overlapping [82]. Also, permuting $p$-values of SNVs in proximity to the baseline model of a gene, can result in discrepancies between $p$-value correlations and true genetic correlations amongst those SNVs, and SNVs from nearby within the baseline model of the gene. Still, we point out that

EPVP represents the first procedure to control genuine augmentations with random counterparts, whilst striving to maintain relevant confounders. Thirdly, without gold-standard lists of phenotype-modifying genes or gene sets derived independently from GWAS, it is problematic to quantify, objectively, to what extent augmentation actually improves sensitivity and specificity of gene-set analyses for GWAS data. That being said, our results strongly stress that evaluating such benefits is not as trivial as comparing between numbers of significant genes or gene sets detected with different mappings.

In conclusion, we have shown that integrating datasets of RIs into gene-set analyses for GWAS data can strengthen the association of biologically-relevant gene sets with phenotypes. We demonstrate that such analyses must include carefully designed controls to reduce spurious discoveries. We believe that the potential benefits of augmentation are, to an extent, curtailed by mismapping, and that the generation of additional cell-type datasets of RIs may help to partially resolve this issue. Our work advances previous efforts to incorporate RIs into gene-set analyses for GWAS data, most importantly by introducing an array of controls to distinguish between genuine improvements, and trivial or confounded ones.

## Methods

### Mapping of SNVs to genes and integration of regulatory interactions

We generated all SNV-to-gene mappings on a background of 19,700 protein-coding genes (human GENCODE release v33 for GRCh37 [83]) and 7,398,358 common (that is, minor-allele frequency ≥ 1%), biallelic and unambiguous (referring to strand-ambiguity) SNVs represented in the European reference-population of the 1000-Genomes Project [84] (specifically, a subset of SNVs from the MAGMA European-population binary files, which are downloadable through the MAGMA website: https://ctg.cncr.nl/software/magma). These counts refer to genes and SNVs located on chromosomes 1–23, after omitting genes and SNVs overlapping with the MHC region (positions 28,477,797–33,448,354 on chromosome 6). The MHC region is routinely removed from GWAS data analyses for reasons related to its unusual LD structure [85]. To clarify, the actual numbers of genes and SNVs analyzed were always less than those reported above, since these were ultimately determined by the presence or absence of SNVs in a GWAS dataset.

We retrieved, and minimally processed, numerous datasets of RIs (Tables 1 and S1) from four categories: (a) computationally predicted EPMs: DHS07 or DNase-hypersensitivity correlations based on ENCODE data (requiring correlation above 0.7 between enhancer and promoter DHS profiles) [local dataset], FOCS [42], GeneHancer [43], JEME [44], and PsychENCODE [45]; (b) HiC datasets: fetal brain [46] and adult brain [45]; (c) pc-HiC datasets: selected tissues and cell types, as well as a global dataset based on 24 tissues and cell types [47]; and (d) cMap datasets (that is, a collection of datasets that were provided by the following resources): brain-neuronal cells [48], heart-muscle cells [49], pancreatic-islet cells [50], prostate-epithelial cells [51], and blood cells [52]. The specific pc-HiC and cMap datasets used for each phenotype are detailed in S1 Table. Processing involved limiting interactions to those implicating genes from our background set of protein-coding genes, retaining only intra-chromosomal interactions, and merging, separately for each gene, any overlapping regulatory elements. Except for the HiC datasets, the downloaded RIs were already characterized as relationships between genomic loci (that is, regulatory elements) and genes they regulate. Using bedtools [86], we defined RIs for the HiC datasets ourselves, by treating a locus that was paired with another locus overlapping a promoter of a particular gene, as a regulatory element for that gene. For this purpose, we treated the 2kb-region upstream of the transcription-start site of a gene, as its promoter.

We built SNV-to-gene mappings by leveraging the MAGMA [31] annotate command. This command takes a file with features and their coordinates (that is, a feature-coordinate file), alongside a file with SNVs and their coordinates, to generate a file defining each feature by SNVs overlapping with it. Normally, a feature-coordinate file (that is, a MAGMA.gene.loc file) describes genes (by their gene symbol) and their coordinates. Instead, we generated feature-coordinate files separately for gene bodies, upstream flanks (2kb, 10kb, 20kb, 35kb, 50kb, 100kb, 250kb, 500kb, and 1000kb), downstream flanks (2kb, 10kb, 20kb, 35kb, 50kb, 100kb, 250kb, 500kb, and 1000kb), and each processed dataset of RIs. To clarify, a feature-coordinate file for a processed dataset of RIs, stored, on each row, the coordinates of a regulatory element for a specific gene. Each regulatory element was named according to the gene it associated with (as in a conventional MAGMA.gene.loc file), but also contained a unique identifier in the name (a digit) to distinguish it from other regulatory elements for the same gene in the dataset. We annotated each feature-coordinate file with SNVs from the MAGMA European-population binary files (specifically, the.bim file) to build diverse SNV-to-feature mappings. We used the following command:

```
magma—annotate
—snp-loc path/to/binary/bim/file
—gene-loc path/to/feature/coordinate/file
—out path/to/output/files/and/prefix
```

Annotating feature-coordinate files independently from each other, while retaining relevant information in the feature name, gave us the flexibility to aggregate resulting SNV-to-feature mappings into a large assortment of SNV-to-gene mappings. For example, the baseline model was built by aggregating, at the level of a gene, the SNV-to-feature mappings of gene bodies, 10kb upstream flanks, and 10kb downstream flanks. We ensured that the same SNV was never mapped more than once to the same gene, but we allowed the same variant to be mapped to more than one gene (like in the case of overlapping gene bodies). Aggregation of mappings was performed in R.

## Procurement and processing of GWAS data

We downloaded GWAS data (summary statistics) from European-cohort studies for ten phenotypes (Table 2) related to various traits and diseases (Alzheimer's disease [53], atrial fibrillation [54], bone density [55], breast cancer [56], coronary-artery disease [57], Crohn's disease [58], macular degeneration [59], prostate cancer [60], schizophrenia [61], and type-2 diabetes [62]). To (i) remove low-quality data, (ii) make data amenable to analysis with MAGMA, and (iii) ensure concordance between variant definitions (that is, guaranteeing that an rs-identifier always signified the same SNV in terms of coordinate and allele information) across summary statistics, SNV-to-gene mappings, and MAGMA European-population binary files, we processed all summary statistics (using R) in the same way.

Briefly, we first filtered summary statistics to retain only biallelic, unambiguous SNVs with valid statistics (effect-size, standard-error, *p*-value and sample-size). Then, we assigned rs-identifiers to SNVs by matching their meta-information (chromosome, position, and alternative alleles), with a dictionary relating such meta-information to rs-identifiers. We created this dictionary from the MAGMA European-population binary files (specifically the.bim file), which were also used to build all SNV-to-gene mappings, and conveniently, related rs-identifiers to meta-information in a one-to-one manner. Any unmatched SNVs were identified through their rs-identifier provided originally in the summary statistics (if provided at all), under the condition that this rs-identifier was represented in our dictionary alongside the same combination of alternative alleles. In these cases, chromosome and position information of SNVs was updated to match the dictionary. We then eliminated any still-unmatched SNVs,

duplicate SNVs (that is, any rs-identifier that occurred more than once with different statistics), rare SNVs (minor-allele frequency < 1% according to the MAGMA European-population binary files), and SNVs overlapping with the MHC region. Finally, we removed SNVs with an outlying sample-size (defined as a sample size more or less than five standard deviations from the mean across remaining SNVs).

## Gene-scoring analysis and gene-set analysis with MAGMA

We used MAGMA (v1.08) to compute unadjusted gene scores (MAGMA's "SNP-Wise Mean" method) from SNV-level associations for selected combinations of phenotypes (that is, processed summary statistics) and SNV-to-gene mappings. For each combination we obtained an output file (MAGMA.raw file) with unadjusted gene scores alongside gene-gene correlations based on LD (determined by MAGMA with the MAGMA European-population binary files). We used the following command:

```
magma—bfile path/to/binary/files
—pval path/to/summary/statistics use = rs_id,pval ncol = sample_size
—gene-annot path/to/snv/to/gene/annot/file
—out path/to/output/files/and/prefix
```

We used each output file (MAGMA.raw file) as input for competitive gene-set analysis with MAGMA, to obtain gene-set scores (P column in MAGMA.gsa.out file) as well as adjusted gene scores (ZSTAT column in MAGMA.gsa.genes.out file). This procedure adjusts gene scores for known confounders (that is, outlying scores are truncated before correcting for residual effects of SNV count, within-gene LD, inverse mean minor-allele count, and sample size), before executing the gene-set analysis proper, using previously computed gene-gene correlations to account for dependencies between genes assigned to the same gene set (see MAGMA's original publication, as well as the user manual downloadable from the MAGMA website, for full details). Gene-set analyses were executed with 2,647 GO biological-process gene sets [63] (based on the c5.bp.v7.1.symbols.gmt file from the GSEA website of the Broad Institute), each of which contained between 25 and 200 genes (before gene-set analysis itself, which removed any additional genes without a score). We ensured that genes overlapping with the MHC region were not used to obtain adjusted gene scores and gene-set scores, by leveraging the gene-exclude modifier under the settings flag. We used the following command:

```
magma—gene-results path/to/magma/raw/file
—set-annot path/to/gene/sets/file
—settings gene-info gene-exclude = MHC_genes.list
—out path/to/output/files/and/prefix
```

Augmentations usually increased the number of genes with a score, because not all baseline models of genes contained even one SNV (also present in a dataset of summary statistics) required for calculating their score. We saw this as a possible advantage of augmentation, and always executed gene-set analyses with the entire collection of scored genes. Still, we restricted our evaluations of genes and gene sets to those with a score for every SNV-to-gene mapping. This ensured that we were always be able to compare the score of a gene or a gene set between all SNV-to-gene mappings, and it overcame confounding when comparing numbers of significant genes or gene sets detected between SNV-to-gene mappings.

Regarding scores, adjusted gene scores were reported as Z-statistics, so we converted them to upper-tail $p$-values before FDR-adjustment for multiple-testing. However, to rank genes for IRED, to visualize IRED results (Δ Gene Score), and to illustrate gene scores in our case studies, we simply used the reported Z-statistics. Gene-set scores were reported as upper-tail $p$-values, and so, used directly in FDR-adjustment for multiple-testing. For the analysis where we validated responses of gene sets to augmentation with the EPVP procedure, we transformed

FDR-adjusted, upper-tail $p$-values to corresponding Z-statistics (by probit transformation of one minus an FDR-adjusted, upper-tail $p$-value), and we used these Z-statistics to compare (as described in the results) between runs with genuine and permuted data. Likewise, for visualizing results from this validation procedure, as well as those from IRED ($\Delta$ Gene-Set Score), we transformed FDR-adjusted upper-tail $p$-values of gene sets to corresponding Z-statistics.

### Limiting redundancy amongst significant gene sets with "rrvgo"

Given the hierarchical structure of the GO ontology, many gene sets show high overlap [64], and thus, significantly enriched gene sets are typically highly redundant. We used the "rrvgo" package in R [65] to eliminate redundant gene sets from the list of significant gene sets identified by MAGMA gene-set analysis for each combination of phenotype and SNV-to-gene mapping. In brief, we first computed the similarity matrix (`calculateSimMatrix` function) for our 2,647 GO biological-process gene sets with the SimRel algorithm (additional arguments: `orgdb = "org.Hs.eg.db"`, `ont = "BP"`, `method = "Rel"`). In calculating the similarity matrix, the majority of our gene sets (that is, 2,625 gene sets) were recognized (Bioconductor version 3.12), so any technical drop-off caused by running rrvgo was negligible. For each combination of phenotype and SNV-to-gene mapping, we then eliminated redundant gene sets from the list of significant gene sets (`reduceSimMatrix` function), using the default stringency setting (`threshold = 0.7`) with -$\log_{10}$ transformed, FDR-adjusted $p$-values for gene-set scores (as recommended).

All our gene-set analyses focused on non-redundant, significant gene sets. In one supplementary analysis (S7 Table), where we counted novel (significant only with the augmented model of interest), known (significant with both the augmented model of interest and the baseline model), and lost (significant with only the baseline model) gene sets, we occasionally restored some redundancy. More precisely, for this analysis we considered the union of non-redundant, significant gene sets identified across the two SNV-to-gene mappings being compared. The reason was that every now-and-then a significant gene set that had been eliminated (for reason of redundancy) for one SNV-to-gene mapping, had not been eliminated for the other. Since such a gene set was, after all, statistically significant with both mappings, we counted it towards the known gene-set category.

### Calculating the coverage of a mapping

The coverage of a mapping was calculated as the number of non-redundant base pairs covered by all features defining a gene, summed across all genes. Note that the coverage of a certain mapping was not necessarily identical across all phenotypes. This stems from the fact that we considered a feature to define a gene not simply based on a SNV-to-gene mapping, but also in a way specific to a phenotype. That is, a feature (a gene body, a flank, or a regulatory element) was said to define a gene if, and only if, it contained at least one SNV that was also represented in a particular dataset of summary statistics (in other words: if, and only if, the feature also contributed towards the score of the gene).

### The EPVP procedure—controlling results with matched, random regulatory interactions or flanks

To investigate the hypothesized benefits of augmenting the default SNV-to-gene mapping scheme of MAGMA with RIs, we defined this default mapping scheme as our baseline model (that is, gene bodies with 10kb upstream- and downstream flanks), and on top of this baseline model, we incorporated information from various sources of RIs. To scrutinize the effects of incorporating such information, we developed a permutation control, which we called

extragenic *p*-value push (EPVP). EPVP yields gene scores and gene-set scores under conditions where SNVs located within regulatory elements or flanks of each gene, but not part of their baseline model definition, are assigned inter-correlated, background SNV-level associations (*p*-values) and sample sizes (in place of the original, or genuine, values).

The EPVP procedure was run separately for each combination of phenotype and augmented SNV-to-gene mapping. First, for a selected combination, we defined the SNVs mapped to each gene according to the augmented SNV-to-gene mapping, and removed any SNVs not represented in the summary statistics. Next, for each gene, we labelled SNVs mapped to it via the baseline model as intragenic, and remaining SNVs (that is, SNVs mapped to it exclusively via the augmentation) as extragenic. Then, we performed 20 permutations in which *p*-values and sample sizes of extragenic SNVs of individual genes, were assigned random, but inter-correlated, background values. Specifically, we arranged all SNVs in the summary statistics numerically along their coordinate positions within each chromosome. Then, for each permutation, SNV-level associations (*p*-values) and associated sample sizes were shifted en masse by a randomly-chosen index within each chromosome. Chromosomes were treated as circular, such that indices surpassing the end of a chromosome, were pushed back to its start. For each gene, we annotated its extragenic SNVs with the permuted *p*-values and sample sizes, and its intragenic SNVs with the original *p*-values and sample sizes. A miniature summary-statistics file was then generated for each gene (that is, a file containing just the SNVs annotated to a gene, alongside their *p*-values and sample sizes), and named according to the gene and the permutation (1–20).

Each miniature summary-statistics file was used separately for gene scoring with MAGMA to obtain 20 unadjusted gene scores for each gene, one for each permutation. MAGMA runs for this procedure were performed like regular runs, except that a separate run was required for each gene and permutation combination. To improve efficiency, we enabled the `-genes-only` flag, which instructed MAGMA to not calculate gene-gene correlations, and in turn, to output unadjusted gene scores in a results file slightly different from the one described above (known as a MAGMA.out file, instead of a MAGMA.raw file). To obtain adjusted gene scores and gene-set scores, we required the file containing gene scores with gene-gene correlations (that is, the MAGMA.raw file) obtained with the regular run (that is, the file obtained when executing gene-scoring analysis with the relevant, complete, and non-permuted dataset of summary statistics, and the relevant, augmented SNV-to-gene mapping). Precisely, for each permutation, we replaced the score of each gene in this file (that is, the 9th column in a MAGMA.raw file) with its corresponding permuted score (that is, the ZSTAT column in a MAGMA.out file). We likewise replaced the sample size associated with each gene (that is, the 7th column in a MAGMA.raw file and the "N" column in a MAGMA.out file), which reflected the sample size amongst SNVs mapped to a gene. Other columns, including gene-gene correlations, were not affected by EPVP, and were thus left unaltered. Ultimately, we generated 20 new MAGMA.raw files, one for each permutation. We used each one to obtain adjusted gene scores and gene-set scores (that is, 20 scores for each gene and gene-set) by executing gene-set analysis as usual.

EPVP keeps the actual SNVs assigned to each gene unchanged, thus preserving the number of SNVs assigned to each gene. Also, by shifting the SNV-associations together within a chromosome, EPVP attempts to maintain LD-related signal correlations (in *p*-values) between SNVs and genes (by shifting only once for a permutation, and not separately for every gene). We permuted the sample size and SNV-level association together for each SNV, because sample size affects the latter. The principles of EPVP were adapted from a circular genomic-permutation approach used to condition SNV-level associations on the background of complete GWAS data (summary statistics), before executing gene scoring and gene-set analysis [87].

More recently, the circular genomic-permutation approach was implemented as a computationally-efficient algorithm (not specifically relevant to our work), and then used to analyze GWAS data (summary statistics) for asthma [88].

We note that EPVP is computationally demanding to execute (both in terms of storage requirements and run time). For this reason, we performed EPVP for only seven selected augmentations for each phenotype.

### Gene scoring analysis with Pascal

We used Pascal [30] to compute gene scores ("Sum" method) from SNV-level associations for each phenotype (that is, processed summary statistics) using a variety of flank-based SNV-to-gene mappings. Unlike MAGMA, Pascal does not accept custom SNV-to-gene mappings, and users can only add flanks on-top-of gene bodies (via the -up = X and -down = Y flags). We used default settings to run Pascal, although we (1) turned-off the maximum limit for the number of SNVs per gene, and we (2) allowed SNVs to have a minor-allele frequency $\geq 1\%$ (instead of $\geq 5\%$). For additional consistency with our MAGMA analyses, we replaced the Pascal reference population binary files with the MAGMA European-population binary files, and we performed gene scoring for the same list of 19,700 protein-coding genes analyzed with MAGMA. We restricted our evaluations (S4 Table) to genes with a score for every SNV-to-gene mapping in both Pascal and MAGMA.

### Supporting information

**S1 Fig. An illustrative description of the EPVP permutation control. (A)** SNVs in a dataset of GWAS summary statistics are arranged along their coordinate positions. For each permutation, SNV-level associations (depicted as varying tones of red) are shifted collectively by a randomly-chosen index within each chromosome (note that, only one chromosome is shown). Chromosomes are treated as circular, such that indices surpassing the end of a chromosome, are pushed back to its start. **(B)** A small region of our chromosome (bordered by a black box in A) is shown in detail. Genes (rectangles) and their regulatory elements (triangles) are illustrated on one track, and SNVs (circles) are repeated across multiple, separate tracks (one for each of the labelled scenarios). For each permutation of EPVP, SNVs located within regulatory elements or flanks of each gene, but not part of their baseline-model definition (that is, our so-called extragenic SNVs), are assigned background SNV-level associations (depicted as circles with a pattern-fill on runs with permuted data). SNV-level associations amongst SNVs mapped to a gene via the baseline model are left unaltered (depicted as circles with a solid, non-white fill). SNVs not mapped to any gene (according to the mapping) are also shown, but do not affect the score of any gene (depicted as circles with a solid, white fill). **(C)** Gene scores (for genes G1, G2, G3, . . .) and gene-set scores (for gene sets S1, S2, S3, . . .) are computed with the baseline model using unpermuted data (Baseline Model in B), with the augmented model using unpermuted data (Augmented Model in B), as well as with the augmented model using permuted data (20 independent permutations of EPVP; Permutation 1–20 in B). Row-wise evaluations focus on the numbers of significant genes or gene sets detected, whereas column-wise evaluations focus on responses of individual genes or gene sets to augmentation. Abbreviations: Chr1 (Chromosome 1); Perm (Permutation); RE (regulatory element); G1 (Gene 1); S1 (Gene Set 1); P1 (Permutation 1).
(PDF)

**S2 Fig. The cumulative distribution of significant genes ranked by their size for selected mappings.** If the size of genes is unrelated to their tendency to be significant, the cumulative

distribution of significant genes ranked by their size is expected to fall approximately on the diagonal. Downward deviation from the diagonal implies that large genes are preferentially significant, while upward deviation from the diagonal implies that small genes are preferentially significant. A deviation from the diagonal is not necessarily a bias, because genes associated with a phenotype might have a genuine tendency to be either large or small (after all, immune-related genes tend to be small and neuron-related genes tend to be large [89]). To evaluate whether or not augmentation results in a confounding gene-size effect (that is, a systematic effect on gene scores that occurs with random augmentation), it is therefore necessary to test for a consistent deviation (that is, consistent across phenotypes and in terms of directionality) from the diagonal that is induced (that is, occurs relative the baseline-model distribution) by random augmentation. We expect at least that the baseline-model distribution does not show evidence for a gene-size effect, since MAGMA's gene scores are said to be well controlled for gene size (see original publication [31] and MAGMA website for details). **(A)** Hypothetical illustration of a gene-size effect where larger genes are preferentially significant not just with genuine augmentation (green), but also with matched, random augmentation (grey; one distribution for each of 20 independent permutations of EPVP). Except for the baseline-model distribution (blue), these distributions are not based on real data and serve only to clarify the previous points. We point out that, as expected, no gene-size effect is observed with the baseline model. In most cases involving real data **(B-E, G, H)**, there was no prominent (that is, observable) relationship between the size of genes and their tendency to be significant. **(F)** However, augmentation with random counterparts to the global pc-HiC dataset of regulatory interactions, almost consistently skewed the distribution to fall below the diagonal, such that large genes were preferentially significant (refer to S3 Fig for clarity). This suggests that augmentation can induce a prominent, non-specific (that is, we are looking here at random augmentation) gene-size effect, in which large genes become preferentially significant over small genes. Given the self-contained nature of gene scores (that is, the score of one gene does not depend on the score of any other gene), and the fact that more heavily-augmented genes are probably enriched amongst larger genes (that is, the right-hand side of the distribution), these data caution, more specifically, that random augmentation of a gene can improve its gene score. Phenotype abbreviations: Alzheimer. (Alzheimer's disease); Atrial Fib. (atrial fibrillation); Bone Den. (bone density); Breast C. (breast cancer); C-Artery D. (coronary-artery disease); Crohn's (Crohn's disease); Mac. Deg. (macular degeneration); Prostate C. (prostate cancer); Schizo. (schizophrenia); T2 Diab. (type-2 diabetes). The augmentation, "Big Flanks", refers to gene bodies with 100kb upstream- and downstream flanks (note, as part of the baseline model, the first 10kb on either side of a gene body were not permuted by EPVP).
(PDF)

**S3 Fig. Random augmentations can occasionally improve gene scores.** The following figures are enlargements of **(A)** S2B Fig and **(B)** S2F Fig, respectively (refer to the caption of S2 Fig for an explanation).
(PDF)

**S4 Fig. Significant gene sets that gain from augmentation, often gain no more from genuine augmentation than from matched, random augmentations.** This is an extension of Fig 6 from the Main Text to all other phenotypes (refer to the caption of Fig 6 for an explanation). **(A)** Alzheimer's Disease, **(B)** Bone Density, **(C)** Breast Cancer, **(D)** C-Artery Disease (Coronary-Artery Disease), **(E)** Crohn's Disease, **(F)** Mac. Degeneration (Macular Degeneration), **(G)** Prostate Cancer, **(H)** Schizophrenia, and **(I)** Type-2 Diabetes.
(PDF)

**S5 Fig.** *Post-synaptic chemical transmission* **is a schizophrenia-associated gene set that demonstrates a robust, validated gain from augmentation with the pc-HiC of brain dataset of regulatory interactions. (A)** A comparison between gene-set scores (that is, each score is based on the probit transformation of one minus the relevant, FDR-adjusted, upper-tail $p$-value) obtained using the baseline model, the baseline model augmented with genuine regulatory interactions, and the baseline model augmented with matched, random regulatory interactions. Bigger, positive scores imply stronger enrichment for phenotype association. For random augmentation, counts and error bars represent the mean and standard deviation, respectively (based on 20 independent permutations of EPVP). Only gene sets detected as statistically significant with genuine augmentation are shown. Red, dashed line shows the significance cut-off ($\alpha = 0.05$). Each gene set was assigned to one of three validation categories to reflect how pronounced a gain was with genuine augmentation over that with matched, random augmentation (see Main Text). **(B)** The gain for the *post-synaptic chemical transmission* gene set was robust (refer to the caption of Fig 7 and the Main Text for an explanation). Top-gaining genes that had to be removed from the gene set for its gain to be lost are labelled. (PDF)

**S6 Fig.** *Endocrine system development* **is a type-2 diabetes associated gene-set that demonstrates a robust, validated gain from augmentation with the pc-HiC of pancreatic-islet cells dataset of regulatory interactions. (A)** A comparison between gene-set scores (that is, each score is based on the probit transformation of one minus the relevant, FDR-adjusted, upper-tail $p$-value) obtained using the baseline model, the baseline model augmented with genuine regulatory interactions, and the baseline model augmented with matched, random regulatory interactions (refer to the caption of S5A Fig for an explanation). **(B)** The gain for the *endocrine system development* gene set was robust (refer to the caption of Fig 7 and the Main Text for an explanation). Top-gaining genes that had to be removed from the gene set for its gain to be lost are labelled. Mapping abbreviations: Pa-Islet Cells (pancreatic-islet cells). (PDF)

**S7 Fig. Case study on the bone-density associated gene-set,** *response to BMP (bone-morphogenetic protein)*, **which demonstrates a robust, validated gain from augmentation with the global pc-HiC dataset of regulatory interactions. (A)** A comparison between gene-set scores (that is, each score is based on the probit transformation of one minus the relevant, FDR-adjusted, upper-tail $p$-value) obtained using the baseline model, the baseline model augmented with genuine regulatory interactions, and the baseline model augmented with matched, random regulatory interactions (refer to the caption of S5A Fig for an explanation). To save space, only a maximum of five gene sets per validation category are shown (selected according to their gene-set score with genuine augmentation). Overall, augmentation of the baseline model with the global pc-HiC dataset of regulatory interactions, led to the detection of 14 gene sets significantly associated with bone density. Only four of these gene sets gained from augmentation, with three of the gains being strongly validated (*response to BMP*, *negative regulation of small GTPase mediated signal transduction*, and *mesenchymal cell proliferation*) and one mildly validated (*regulation of lamellipodium organization*) by the EPVP procedure. **(B)** The strongly validated gain for the gene set, *response to BMP* (officially, *go_response_to_BMP*), which is relevant to bone formation [90] and contained 171 genes (note that, 153 of these genes had a score with both models, and that the other 18 genes did not have a score with either model), was the only robust gain (refer to the caption of Fig 7 and to the Main Text for an explanation) (S11 Table). More precisely, the gain for this gene set was lost only after removal of the 14 top-gaining genes from the gene set (namely and in order, *TGFB2*, *SOX9*, *PELO*, *PPM1A*, *GDF6*, *GDF1*, *AMH*, *SMPD3*, *SMAD2*, *WNT5A*, *NOG*, *HTRA1*, *FKBP8*, and *SULF1*). **(C)** For many of

the genes mentioned, we were able to identify interesting regulatory elements located in bone-density associated loci, and to additionally support their link to the relevant genes with eQTL data (refer to the caption of Fig 8 for an explanation). For example, *AMH* gained from eQTL-supported regulatory elements located both upstream (~100kb, ~250kb and ~650kb) and downstream (~300kb and ~500kb) from the transcription-start site of the gene. Serum levels of *AMH* are associated with bone density in premenopausal women with ovarian insufficiency [91].
(PDF)

**S8 Fig. Case study on the prostate-cancer associated gene-set, *regulation of embryonic development*, which demonstrates a robust, validated gain from augmentation with the EPM of GeneHancer dataset of regulatory interactions. (A)** A comparison between gene-set scores (that is, each score is based on the probit transformation of one minus the relevant, FDR-adjusted, upper-tail *p*-value) obtained using the baseline model, the baseline model augmented with genuine regulatory interactions, and the baseline model augmented with matched, random regulatory interactions (refer to the caption of S5A Fig for an explanation). Overall, augmentation of the baseline model with the EPM of GeneHancer dataset of regulatory interactions, led to the detection of eight gene sets significantly associated with prostate cancer. Seven of these gene sets gained from augmentation, and amongst them, three gains were strongly validated, two gains were mildly validated, and two gains were invalidated, by the EPVP procedure. **(B)** Gains were robust only for the three strongly validated gene sets, including the *regulation of embryonic development* (officially, *go_regulation_of_embryonic_development*) gene set (refer to the caption of Fig 7 and to the Main Text for an explanation) (S11 Table). This gene set, which contained 133 genes (note that, 127 of these genes had a score with both models, and that the other six genes did not have a score with either model), required the removal of the 13 top-gaining genes from the gene set (namely and in order, *PHLDA2*, *TRAF3IP1*, *LAMA5*, *HNF4A*, *BMP4*, *LFNG*, *FGFR1*, *TGIF2*, *WDPCP*, *CCSAP*, *TBX2*, *MBP*, and *STK4*) before the gain for the gene set itself was lost. Though the biological relevance of this gene set is not be directly obvious, it may reflect the role of basic cellular processes (growth, differentiation and movement) in cancer [92]. **(C)** Remarkable gains from augmentation were observed for three of the genes mentioned (*PHLDA2* via an eQTL-supported regulatory element ~750kb downstream from the transcription start-site; *TRAF3IP1* via regulatory elements ~900kb upstream from the transcription start-site; *LAMA5* via regulatory elements ~1400kb upstream from the transcription start-site) (refer to the caption of Fig 8 for an explanation). It is noteworthy that *LAMA5* has formerly been associated with colorectal cancer [93].
(PDF)

**S1 Table. Overview of pc-HiC and cMap datasets selected for each phenotype.**
(DOCX)

**S2 Table. Number of significant genes detected using baseline model without and with augmentation.**
(DOCX)

**S3 Table. Number of novel, known, and lost, significant genes resulting from augmentation of the baseline model.**
(DOCX)

**S4 Table. Significant genes detected with MAGMA's unadjusted- and Pascal's gene-scores.**
(DOCX)

**S5 Table. Number of significant genes detected using baseline model without and with augmentation including EPVP.**
(DOCX)

**S6 Table. Number of significant gene sets detected using baseline model without and with augmentation.**
(DOCX)

**S7 Table. Number of novel, known, and lost, significant gene sets resulting from augmentation of the baseline model.**
(DOCX)

**S8 Table. Number of gaining and non-gaining gene sets amongst significant gene sets detected with augmentation of the baseline model.**
(DOCX)

**S9 Table. Number of strongly validated, mildly validated and invalidated gains amongst significant gene sets that gained from augmentation of the baseline model.**
(DOCX)

**S10 Table. Number of robust and non-robust gains amongst significant gene sets demonstrating validated (mildly and strongly) gains from augmentation of the baseline model.**
(DOCX)

**S11 Table. Listing of IRED results (for robustness) amongst gene sets demonstrating validated (mildly and strongly) gains from augmentation of the baseline model.**
(XLSX)

## Author Contributions

**Conceptualization:** David Groenewoud, Avinoam Shye, Ran Elkon.

**Formal analysis:** David Groenewoud, Avinoam Shye.

**Investigation:** David Groenewoud, Avinoam Shye.

**Methodology:** David Groenewoud, Avinoam Shye.

**Supervision:** Ran Elkon.

**Visualization:** David Groenewoud, Ran Elkon.

**Writing – original draft:** David Groenewoud, Ran Elkon.

**Writing – review & editing:** David Groenewoud, Ran Elkon.

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
