## [Decision Letter · Decision Letter 0]

2 Nov 2021

Dear Dr. Elkon,

Thank you very much for submitting your manuscript "Incorporating regulatory interactions into gene-set analyses for GWAS data: a controlled analysis with the MAGMA tool" for consideration at PLOS Computational Biology.

As with all papers reviewed by the journal, your manuscript was reviewed by members of the editorial board and by several independent reviewers. In light of the reviews (below this email), we would like to invite the resubmission of a significantly-revised version that takes into account the reviewers' comments.

We cannot make any decision about publication until we have seen the revised manuscript and your response to the reviewers' comments. Your revised manuscript is also likely to be sent to reviewers for further evaluation.

Sincerely,

Saurabh Sinha

Guest Editor

PLOS Computational Biology

Sushmita Roy

Deputy Editor

PLOS Computational Biology

Reviewer's Responses to Questions

**Comments to the Authors:**

Reviewer #1: The manuscripts seeks to systematically evaluate the effect of different types of augmentation (based on flanks and based on regulatory interactions / RIs) on SNV-to-gene mapping and gene set characterization performed for GWAS study. The authors consider multiple GWAS studies and multiple RI datasets to assess this. The design of the study overall is well done and the study is interesting and insightful. I have some comments that I hope can strengthen the study.

Comments:

1- The main goal of the paper is to assess the effect of augmenting the GWAS gene set enrichment analysis with regulatory interactions. In the introduction, the authors talk about different methods that achieve this goal using different strategies. However, as they explain the results, it is unclear what augmentation method they use. For example, the second subsection of Results discusses the effect of augmentation on number of significant genes identified, yet I could not find what method was used to incorporate RIs to augment. While "Methods" describes some details, it is still important to conceptually describe how RIs are used to augment the analysis. Is this augmentation similar to H-MAGMA, similar to e-MAGMA or some other approach? If those above, still more information is needed to describe how this RI augmentation is performed and what it corresponds to.

2- While interesting observations are made, many of them are not quantified (e.g., using statistical tests). It would be necessary to support observations with statistical tests. In following comments I will mention some examples, but the same idea should be applied to every observation/analysis/figure.

3- "For instance, the fetal-brain HiC 240 dataset of RIs, outperformed most other datasets of RIs, even for non-neurological phenotypes. “ Is “outperformed” the right term? Does a larger number of genes represent a higher performance? Why? Just because more significant genes are identified, there is no guarantee that more True Positive genes are identified. In fact, in "Discussion" the authors bring up the possibility of mismatching. So maybe this should be reworded.

4- This comment relates to my first comment above related to lack of description of RI augmentation: depending on how this augmentation is done, the size of RI datasets, the number of interactions for each gene, etc. may affect the number of significant genes they identify in “Gene Scoring Analyses”. The authors have evaluated one such important confounder (i.e., the coverage defined as the number of non-redundant base pairs covered by all features defining a gene, summed across all genes). This analysis is well supported by the trend in Table S2B and Figure S1A. Depending on the method of augmentation, other factors may also be important (e.g., number of interactions, etc.). 

5- The caption of supplementary tables (and figures) should be expanded. They are too short and do not have sufficient information for one to understand them without checking various places in the paper, making it difficult to read.

6- Figure 1B: Based on my understanding, each circle shows a different gene here; in addition, if more points are above the diagonal solid line, it means that larger flank has increased gene-specific scores. If that is the case, please add this description to the caption (and do the same for other figures too). Overall, the captions can be improved for all figures. Also, how about a statistical test (say one-sided Wilcoxon signed rank test) to statistically show whether there is a significant increase in the gene scores when flank size is increased from 10kb to 100kb.

7- Figure 1A: what is the correlation coefficient and its p-value? In general, it is necessary to quantify observations (such as this one) in the manuscript. I will not repeat my suggestions for every figure, but please add quantification in various places (it is right now lacking).

8- While the permutation tests (EPVP) support that not “any” increase in the coverage would result in higher gene scores, the strong trend observed in Fig 1A and table S2B is still concerning and may suggest that this permutation test is not sufficient to rule out the possibility that size of gene region (i.g., gene body + flanks or gene body + RI regions) is the major driver in the number of significant genes. For example, if one keeps increasing the size of the flanks, well beyond 100kb, does the trend remain? At some point, we should expect that a very large flank would not be meaningful; so if the number of significant genes keeps increasing even for very large flanks, that would be concerning.

9- Figure 4 (and other figures): please perform statistical tests to quantify your trends and observations. E.g., in Figure 4B, you can use paired Wilcoxon signed rank test.

10- Discussion: I enjoyed reading the final points about potential for mismatching. Is there other analysis you can use to test for that, since EPVP cannot? Could my suggestion above (increasing flanks sizes beyond reasonable) can (at least indirectly) provide evidence on this hypothesis?

Reviewer #2: Groenewoud et al. investigated how incorporating distal regulatory elements in gene-set based functional analysis of GWAS variants affects the numbers and identities of genes and gene-sets detected as statistically significant. They found that this approach can detect more genes than the current approach but also runs the risk of suggesting spurious associations. To overcome this issue, the authors suggest a set of additional statistical tests.

It is a well written paper. We give several suggestions below. As the authors will note, our main concern is a lack of clarity.

1. To make the paper self-contained, the concepts of competitive/comparative/non-comparative tests need more elaboration. I think, the authors might consider removing these points from the Introduction. The authors also need to clarify what they mean in the Introduction by “expression of gene score”.

2. An easy-to-follow example and figure describing the gene-set analyses tools would be more helpful than the generic description starting with “Gene-set analyses surmise that the collective impact …”.

3. Why did the authors choose MAGMA as the baseline model? This point needs a clear explanation. To what extent are the observations sensitive to the choice of MAGMA vs. other gene-set analysis models they mentioned (PASCAL and VEGAS)?

a. The authors need to better explain MAGMA’s different steps like the “SNP-wise mean” or “adjustment for confounders” in the section titled “Gene Scoring Analyses”.

4. Can the authors comment to what extent the reports of e-MAGMA and H-MAGMA might be spurious (citations #40, #41)?

5. It is interesting how the fetal brain Hi-C RI dataset outperformed other datasets for non-neurological phenotypes. Can the authors offer any explanation? There seems to be a missed opportunity here to elaborate more on the potential caveats of “blindly” incorporating distal variants. Would other filters, such as cell/tissue-type specific accessibility help here? Or, is there an interesting similarity of regulatory mechanism between fetal brains and other organs?

6. Definition of “coverage” – what do the authors mean by “the features defining a gene”?

7. The EPVP statistic is proposed to detect cases “not specific to the genuine SNV level associations, or GWAS p-values, of added SNVs”. However, the initial descriptions do not clarify how the original method takes GWAS p-values and sample-sizes into account. The EPVP statistic also needs more elaboration. Fig. 2 might be more effective with a numeric toy-example. The authors cited other papers (citations #87, #88), but the presented description is unclear in many places. Under the approach of circular shift, can it happen that the number of extragenic variants assigned to a gene actually decreases in some permutations? Due to this lack of clarity, it was not possible for this reviewer to comprehend statements like “flanks and regulatory elements to be located in parts of the genome that are generally enriched for strong phenotype associations; two variables decoupled from each other by EPVP”.

8. The authors mention that EPVP “implied that improvement to gene scores with augmentation could occur randomly”, but did not discuss a false discovery control procedure for gene scores under augmentation approach. For gene-set analysis, however, they have shown how EPVP can invalidate certain gene sets.

9. The IRED results should be presented as main figures.

10. The Discussion section has a candid discussion on the potential limitations of EPVP. These points should be mentioned, at least briefly, upfront.

Minor comments.

1. Can the authors comment whether computational or experimental RIs are more likely to introduce more spurious signals?

2. Some titles/headings are confusing: “Fig. 4. Mappings associated with more significant genes were associated with fewer significant gene sets.”

**Have the authors made all data and (if applicable) computational code underlying the findings in their manuscript fully available?**

Reviewer #1: Yes

Reviewer #2: **No: **The authors did not make the codes available.

PLOS authors have the option to publish the peer review history of their article (what does this mean?). If published, this will include your full peer review and any attached files.

Reviewer #1: No

Reviewer #2: No
---

## [Decision Letter · Decision Letter 1]

9 Feb 2022

Dear Dr. Elkon,

We are pleased to inform you that your manuscript 'Incorporating regulatory interactions into gene-set analyses for GWAS data: a controlled analysis with the MAGMA tool' has been provisionally accepted for publication in PLOS Computational Biology.

Please note that your manuscript will not be scheduled for publication until you have made the required changes, so a swift response is appreciated. Please also note that we expect you will make the code of your method available publicly, as required by the PLOS data policy (see below).

Best regards,

Saurabh Sinha

Guest Editor

PLOS Computational Biology

Sushmita Roy

Deputy Editor

PLOS Computational Biology

Reviewer's Responses to Questions

**Comments to the Authors:**

Reviewer #1: The authors have addressed my comments.

Reviewer #2: None

**Have the authors made all data and (if applicable) computational code underlying the findings in their manuscript fully available?**

Reviewer #1: None

Reviewer #2: **No: **The authors committed to make their codebase public upon acceptance.

PLOS authors have the option to publish the peer review history of their article (what does this mean?). If published, this will include your full peer review and any attached files.

Reviewer #1: No

Reviewer #2: No

---

## [Editor Report · Acceptance letter]

28 Feb 2022

PCOMPBIOL-D-21-01471R1 

Incorporating regulatory interactions into gene-set analyses for GWAS data: a controlled analysis with the MAGMA tool

Dear Dr Elkon,

I am pleased to inform you that your manuscript has been formally accepted for publication in PLOS Computational Biology. Your manuscript is now with our production department and you will be notified of the publication date in due course.

With kind regards,

Orsolya Voros
